

# Impacts of Atmospheric Dynamics on Sea-Ice and Snow Thickness at a Coastal Site in East Antarctica

Diana Francis[1*], Ricardo Fonseca[1], Narendra Nelli[1], Petra Heil[2,3,4] Jonathan D. Wille[5], Irina V. Gorodetskaya[6], Robert A. Massom[2,3,7]

[1] Environmental and Geophysical Sciences (ENGEOS) Lab, Earth Sciences Department, Khalifa University, Abu Dhabi, 127788, United Arab Emirates

[2] Australian Antarctic Division, Department of Climate Change, Energy, the Environment and Water, Kingston, Tasmania, Australia

[3] Australian Antarctic Program Partnership, Institute for Marine and Antarctic Studies, University of Tasmania, Hobart, Tasmania, Australia

[4] Institute Snow and Avalanche Research, Swiss Federal Institute for Forest, Snow and Landscape Research, Davos, Switzerland

[5] Institute for Atmospheric and Climate Science, ETH Zurich, Zurich, Switzerland

[6] Interdisciplinary Centre of Marine and Environmental Research, University of Porto, Porto, Portugal

[7] The Australian Centre for Excellence in Antarctic Science, University of Tasmania, Hobart, Tasmania, Australia

*Correspondence to: diana.francis@ku.ac.ae

**Abstract:**

Antarctic sea ice and its snow cover play a pivotal role in regulating the global climate system. Understanding the intricate interplay between atmospheric dynamics, ocean circulation and mixed-layer properties, and sea ice is essential for predicting future climate change scenarios. This study investigates the relationship between atmospheric conditions and sea-ice and snow characteristics at a coastal East Antarctic site using *in situ* measurements from the winter-spring of 2022. Congruent with previous studies, the observed sea-ice thickness (SIT) follows the seasonal solar cycle with only minor deviations, while the snow thickness variability corresponds closely to cyclonic atmospheric forcing, with significant contributions from katabatic flows and atmospheric rivers (ARs). The *in-situ* measurements highlight the substantial effects of warm and moist air intrusions on the sea-ice, snow and atmospheric state. A high-resolution simulation with the Polar Weather Research and Forecasting model for the 14 November AR highlights the effects of the katabatic winds in slowing down the low-latitude air masses as they approach the Antarctica coastline, with the resulting low-level convergence leading to precipitation rates above $3\,\mathrm{mm\,hr^{-1}}$. Including the observed sea-ice extent and a realistic SIT in the model does not yield more skillful





predictions of surface/near-surface variables and atmospheric profiles. This suggests other factors
such as boundary-layer dynamics and/or land/ice processes may play a more important role than
sea-ice concentration and thickness during AR events. Our findings contribute to a better
understanding of the complex interactions within the Antarctic system, providing valuable insights
for climate modeling and future predictions.
**Keywords:**
Sea Ice, Snow Thickness, SIMBA, PolarWRF, Atmospheric River, Katabatic winds, Antartica

## 1. Introduction

Sea ice, which forms from the freezing of seawater and covers 3-6% of the total surface area
of the planet (depending on season), plays multiple crucial roles in the Earth's climate system and
high-latitude ecosystems (Thomas, 2017; Eayrs et al., 2019). Changes in the formation and melt
rates, extent, seasonality and thickness of Antarctic sea ice - both in the form of drifting pack ice
and less extensive stationary near-shore landfast ice (fast ice) attached to coastal margins and
grounded icebergs (Fraser et al., 2023) - substantially impact the heat and salinity content of the
ocean, and hence the oceanic circulation (e.g., Haumann et al., 2016; Li and Fedorov, 2021). At
the same time, breaks in the sea ice such as leads and recurrent and persistent polynyas (Barber
and Massom, 2007; Francis et al. 2019, 2020; Fonseca et al., 2023) act as a thermal forcing, with
the exposure of ice-free ocean water leading to sensible heat fluxes that can exceed $2000\,\mathrm{W\,m^{-2}}$
and heat up the atmosphere aloft (Guest, 2021), directly impacting the atmospheric flow (Trusel
et al., 2023; Zhang and Screen, 2021). Both oceanic and atmospheric forcing directly impact sea
ice and its spatial extent, seasonality and thickness (Wang et al., 2020; Yang et al., 2021), within
a finely-coupled interactive ocean-sea ice-atmosphere system.
Moreover, sea ice accumulates a highly reflective (high-albedo) and -insulative snow cover
that then strongly modulates the physical and optical properties of the ice cover while also
influencing its formation and melt rates (Sturm and Massom, 2017, and references therein).
Decreases in the thickness and distribution of Antarctic sea ice and its snow cover have strong
potential to impact low-latitude weather patterns (England et al., 2020), disrupt the global surface
energy balance (Riihelä et al., 2021) and amplify climate warming at high southern latitudes
(Williams et al., 2023) - leading to further sea-ice loss that is likely to be further accelerated by
poorly-understood ocean-ice-snow-atmosphere feedback mechanisms (Goosse et al., 2018).
Here, we investigate the impact of atmospheric dynamics on variability in both sea-ice
thickness (SIT) and snow thickness (ST) state through analysis of high-resolution *in-situ*
measurements obtained by an autonomous Snow Ice Mass Balance Array (SIMBA) buoy (Jackson
et al., 2013), combined with atmospheric reanalysis and modeling products. The SIMBA buoy was
deployed from July to November 2022 at a coastal fast-ice site close to Mawson Station in East





Antarctica (at 67.5912°S, 62.8563°E), which will be denoted as "Khalifa SIMBA site on fast-ice
off the Mawson Station" throughout the manuscript. The overall aim of this study is to further our
understanding of the temporal evolution of the thickness and the vertical structure of coastal sea
ice and its snow cover around East Antarctica, and over a six-month period spanning austral winter
through early summer. The motivation is to provide new observations and process information that
will aid numerical-modelling efforts to more accurately simulate the annual cycle of Southern
Ocean sea ice, and observed trends and variability in its distribution (and ultimately thickness) (c.f.
Eayrs et al., 2019). Such an advance is crucial to helping rectify present low confidence in model
projections of future climate and Antarctic sea-ice conditions, that currently diverge for different
models and scenarios (Roach et al., 2020). This study is also particularly timely, given the
precipitous downward trend in Antarctic sea ice extent (SIE) since 2016 (Parkinson, 2019), an
extraordinary record-low annual minimum in February 2023 and a sudden departure to major sea-
ice deficits through the winters of 2023 and 2024 (Reid et al., 2024). This turn of events suggests
that Antarctic sea ice has abruptly shifted into a new low-extent regime (Purich and Doddridge,
2023; Hobbs et al., 2024) due to complex changes in the coupled ocean-ice-snow-atmosphere
system that are far from well understood.
In particular, we here focus on assessing the influence on the sea ice-snow system of: (1) strong
katabatic winds that cascade seawards off the ice sheet and promote sublimation of the sea ice and
its snow cover (Elvidge et al., 2020; Francis et al., 2023); and (2) a number of more ephemeral but
influential extreme atmospheric events in the form of atmospheric rivers (ARs). An AR is a narrow
and highly elongated band of moisture-rich air that originates in the tropics and subtropics and
propagates polewards into the mid- and high-latitudes (Wille et al., 2019; Gorodetskaya et al.,
2020). ARs are associated with increased humidity and cloudiness, leading to an enhancement of
the downward longwave radiation flux while still allowing some of the Sun's shortwave radiation
to reach the surface (Djouma and Holland, 2021). The resulting increase in the surface net radiation
flux gives a warming tendency and promotes surface melting (Gorodetskaya et al., 2013; Francis
et al., 2020; Ghiz et al., 2021). There are several examples of ARs triggering ice and snow melt
around Antarctica e.g., in the Weddell Sea in 1973 and 2017 (Francis et al., 2020); off the Antarctic
Peninsula in March 2015 (Bozkurt et al., 2008) and February 2022 (Gorodetskaya et al., 2023);
around the Amery Ice Shelf in September 2019 (Francis et al., 2021), in West Antarctica (Francis
et al., 2023); and in the Ross Sea (Fonseca et al., 2023).
The recent study of Liang et al. (2023) highlights that the largest impact of ARs on sea ice is
found on the marginal ice zone e.g., a sea-ice extent reduction there that may exceed 10% day[-1].
Reduced coastal offshore SIE may also foster a deeper penetration of the low-latitude air onto the
inland ice sheet, as was the case in the March 2022 "heat wave" in East Antarctica (Wille et al.,
2024a,b). While ARs themselves are relatively rare and short-lived in coastal Antarctica, with a
frequency of ~3 days year[-1] at any given location, the warm and moist air masses they transport
can make a substantial contribution to the surface mass balance (SMB), and they are linked to





extreme precipitation events (Massom et al., 2004; Wille et al., 2021). In East Antarctica, a series of ARs delivered an estimated 44% of the total mean-annual snow accumulation to the high interior ice sheet (in the vicinity of Dome C) over an 18-day period in the austral summer of 2001/2 (Massom et al., 2004), and AR-associated rainfall has exceeded 30% of the total annual precipitation (Mclenann et al., 2022). Moreover, and on Mac. Robertson Land (also in East Antarctica), which includes the Amery Ice Shelf and is the focal region of this study, more than half of the annual precipitation has been observed to fall in the 10 days of heaviest precipitation (Turner et al., 2019). This region also has some of the largest positive trends in AR frequency and AR-related snowfall occurrence in the period 1980-2018. These studies highlight the important impacts of extreme weather events on the coupled Antarctic ocean-ice-snow-atmosphere system, and stresses the need to better understand the role of low-latitude air incursions on the mass balance and state of both the Antarctic Ice Sheet and its surrounding sea-ice cover - and how these may change in a warming climate.

Continuous monitoring since 1978 of the circum-Antarctic spatial extent, concentration and seasonality of sea ice by satellite passive-microwave remote sensing (Parkinson, 2019) has revealed major losses around the continent since 2016 - not only in summer but also latterly through winter (Reid et al., 2024) and for reasons that are not fully understood. This abrupt and precipitous decline has been viewed as a possible regime shift in the coupled ocean-sea ice-atmosphere system (Hobbs et al., 2024). Much less well known - though no less important - are the thicknesses of the ice and its snow cover and whether these are changing. Obtaining more accurate and complete information on the thickness distributions of Antarctic sea ice and its snow cover (and precipitation rates) - and the factors and processes controlling them - is a critical high priority in climate science, particularly in light of climate change (and variability) (Webster et al., 2018; Meredith et al., 2021).

Accurate knowledge of SIT, SIE and concentration is needed to estimate sea-ice volume, a field that is more sensitive to climate change than SIE and SIT alone (Liu et al., 2020) and is also directly parameterized in numerical models (Massonnet et al., 2013; Zhang, 2014; Schroeter and Sandery, 2022). For climate modeling, sea-ice volume (modulated by ST) represents a key integrated measure of the total salinity and freshwater fluxes to the ocean in winter and summer, respectively, and total heat flux to the atmosphere. Current large uncertainties in these quantities prevent proper model evaluation and undermine confidence in model predictions of future Antarctic sea-ice conditions and global weather and climate (Maksym et al., 2012). An analysis of 10 models in the Coupled Model Intercomparison Project Phase 5 (CMIP5) revealed that, around the outer sea-ice zone, changes in sea-ice volume are largely driven by dynamic (wind-driven motion) processes during annual advance and thermodynamic (freeze and melt) processes during the retreat phase, while thermodynamic processes predominate deeper within the sea-ice zone (Schroeter et al., 2018). However, and for the trends, both dynamic and thermodynamic processes are at play,





highlighting the sensitivity of sea-ice volume to changes in oceanic and atmospheric properties
and circulation in response to anthropogenic forcing (Schroeter et al., 2018) and natural variability.

In addition to SIT, reliable large-scale information on the coincident ratio of snow-to-sea ice
thickness is required to determine the distribution of "snow ice" formation around Antarctica
(Maksym and Markus, 2008). By this process, and where the snow is sufficiently thick to depress
the sea-ice surface to below sea level, resultant flooding of the snow creates a slush layer that
subsequently freezes onto the ice surface (Jeffries et al., 1998; Massom et al., 2001). In this way,
snow makes a direct contribution to the sea-mass balance in the freezing season - in addition to its
indirect contribution as a high-albedo insulative layer that moderates Antarctic sea-ice formation
and melt rates (Sturm and Massom, 2017). These factors further underline the need for additional
more accurate information on precipitation and accumulation rates over the sea-ice zone, including
rainfall events (Webster et al., 2018).

Satellite radar and laser altimeters hold the key to large-scale estimation and monitoring of both
SIT (e.g., Fons et al., 2023) and ST (Kacimi and Kwok, 2020). Kurtz and Markus (2012) used the
measurements collected by the Ice, Cloud, and land Elevation Satellite (ICESat) to estimate the
ice thickness around Antarctica. A comparison with ship-based observations revealed a mean
difference of 0.15 m for the period 2003-2008, with a typical SIT of 1-1.5 m. Kacimi and Kwok
(2020), using both laser (ICESat-2) and radar (CryoSat-2) altimeter estimates for the period 1 April
to 16 November 2019, found the thickest sea ice in the western Weddell Sea sector (predominantly
multi-year sea ice), with a mean thickness of 2 m, and the thinnest ice around polynyas in the Ross
Sea and off the Ronne Ice Shelf. Coincident use of laser and radar altimetry also enables basin-
scale estimates of ST. The thickest snow was again observed in the western Weddell Sea
($22.8 \pm 12.4$ cm in May) and the coastal region of the Amundsen-Bellingshausen seas sector
($31.4 \pm 23.1$ cm in September), while the thinnest was in the Ross Sea ($7.35 \pm 4.30$ cm in April)
and the eastern Weddell Sea ($8.21 \pm 5.81$ cm in June) (Kacimi and Kwok, 2020). The studies
mentioned above focus on pack ice, but a similar range of values has been estimated for the
thickness of fast-ice, such as off the Mawson Station (Li et al., 2022) and off the Davis Station
(Heil, 2006) in East Antarctica. Validation of these and other satellite derived estimates of SIT, ST
and sea-ice volume is a crucially important step towards improving their accuracy, yet remains a
considerable challenge, given the lack of regionally- and seasonally-diverse *in situ* and near-
surface observations with which to assess the satellite datasets (Kacimi and Kwok, 2020).

The SIMBA buoy provides high-resolution measurements at a given location of the vertical
temperature profile through the air-snow-ice-upper ocean column, from which snow and ice
thickness can be derived and monitored (Jackson et al., 2013). Time series of such point
observations provide invaluable gap-filling information on the temporal evolution and state of the
snow-sea ice system and its response to atmospheric and oceanic variability. They also provide
crucial information with which to both (i) calibrate the key satellite SIT and ST data products and





(ii) evaluate and improve numerical idealized column and weather forecasting models (Hu et al.,
2023; Plante et al., 2024; Sledd et al., 2024; Wang et al., 2024).

This paper is structured as follows. The observational datasets and model outputs and products
considered, and analysis techniques used, are described in Section 2. The measurements of SIT
and ST, including their variability and the mechanisms behind them, are discussed in Section 3.
Section 4 provides a case-study analysis of the period 11-16 November 2022, while in Section 5
the main findings of the work are outlined and discussed.

## 201    2. Methodology & Diagnostics

In this section, the datasets, model and diagnostics used in this study are described.

### 203    2.1. *In Situ* Measurements at Khalifa SIMBA site off the Mawson Station

*In situ* measurements of SIT and ST are obtained using a sea-ice mass-balance [SIMBA] unit
(Jackson et al., 2013). This SIMBA was deployed on landfast ice offshore from Mawson Station
at 67.5912°S, 62.8563°E (Fig. 1c) on 07 July 2022, and remained *in situ* until 7 December 2022.
The SIMBA unit consists of a 5 m-long thermistor string with a 0.02 m sensors' spacing, a
barometer for surface air pressure, and an external sensor for near-surface ambient air temperature
(Jackson et al., 2013). During deployment, manual measurements of SIT and ST, as well as
freeboard, were recorded. The positions of the sensors relative to the interfaces were noted to
establish the initial state (on 7 July 2022). The measured SIT upon deployment was 0.988 m, the
ST on top of the sea ice was 0.156 m, and the sea-ice freeboard was 0.046 m.

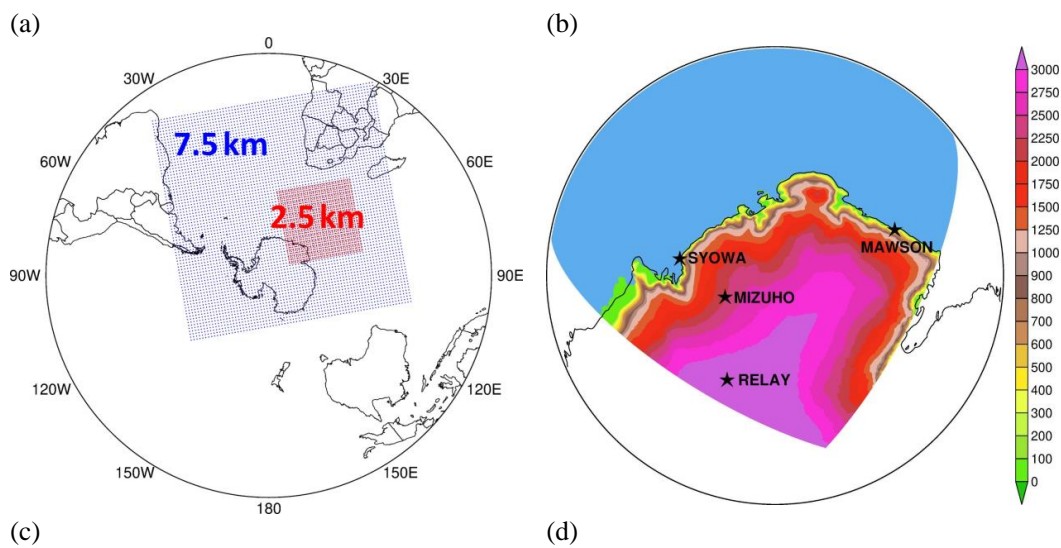





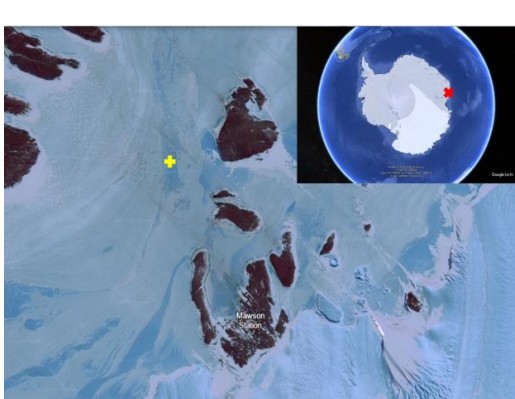
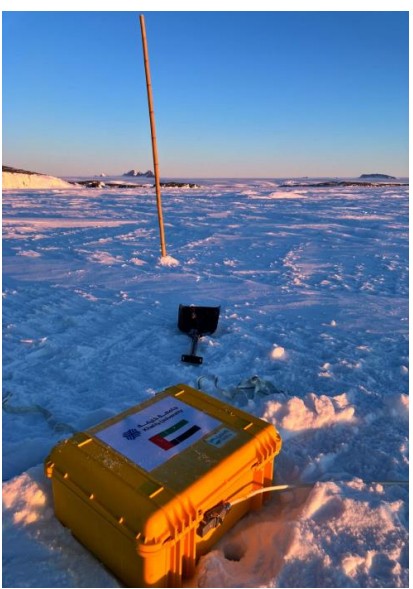

**Figure 1: PolarWRF Simulation:** (a) Spatial extent of the 7.5 km (blue) and 2.5 km (red) PWRF grids used in the numerical simulations. (b) Zoom-in view around East Antarctica for the 2.5 km grid, with the location of the Mawson, Relay, Mizuho and Syowa weather stations highlighted by the stars. The shading gives the orography (m) as seen by the model. (c) SIMBA deployment site (yellow cross) on the fast ice about 1.8 km off Mawson Station. Image source: Landsat 8 acquired on 19 November 2022. The red cross in the inset image, taken from Google Earth Pro, shows where the Mawson Station is located in Antarctica. (d) SIMBA instrument prior to deployment. Image credit: Peter Caithness.



The accuracy of the bus-addressable digital temperature sensing integrated circuit is ±0.0625°C.
A resistor is mounted directly underneath each thermistor sensor. A low voltage supply (8 V) is
connected to each sensor, to gently heat the sensor and its immediate surroundings. In this study,
heating is applied to the sensor chain for durations of 30 s and 120 s once per day, with four vertical
temperature profiles without heating also recorded daily. In this study, SIMBA data from 08 July
to 30 November 2022 is used to assess the evolution of SIT and ST at the site. The measurements
are shown in Fig. 2. For the sensors 6 through 126, the actual temperature and temperature rise
after 120 s heating are given in Fig. 2a and 2b, respectively, with Fig. 2c showing the difference
between the two adjacent temperature sensors after applying the heating. The vertical temperature
gradients in the air above the surface and in the water below the ice bottom are generally very
small (Jackson et al., 2013; Hoppmann et al., 2015; Liao et al., 2018). After 120 s of heating, the
rise in temperature is approximately 10 times higher in air than it is in ice and water (Jackson et
al., 2013). For any two adjacent sensors in the ice, and following the algorithm detailed in Liao et
al. (2018) based on a physical model applied to the SIMBA measurements, the temperature
difference should be ≤ 0.1875°C, whereas for two adjacent sensors in snow, the temperature
difference should be ≥ 0.4375°C. These thresholds are applied to the temperature differences



between adjacent sensors in the heating profile to identify air-snow and snow-ice interfaces
(Jackson et al., 2013; Hoppmann et al., 2015; Liao et al., 2018). The ice-water interface is
identified using a statistical approach based on Liao et al. (2018). A section of the thermistor string,
spanning from the top of the sea ice to a few sensors below the water, is selected. The seawater
temperature near the ice bottom remains stable around the freezing point ($T_f$). The temperature
readings from this section are analyzed as a time series, and the most frequent value is identified
as $T_f$. Scanning from bottom up, the last sensor close to $T_f$ is identified as the ice bottom. The
allowed temperature difference is 1.5 times the thermistor resolution of 0.0625 K. Temporal
evolutions of the three interface locations are plotted in Figs. 2a-c.

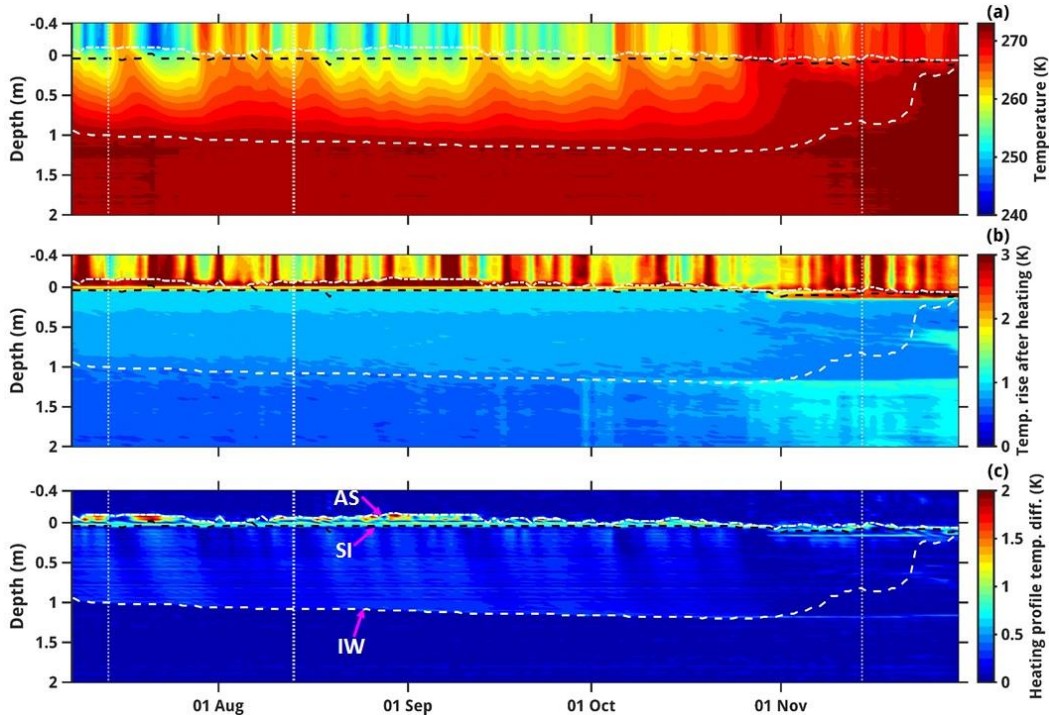

**Figure 2: SIMBA measurements:** (a) Temperature (K) evolution from the top of the chain through the
ice down into the water (the zero line on the y-axis is at the snow-ice interface). (b) Temperature rise (K)
after heating for 120 s. (c) Temperature difference (K) between adjacent sensors after applying the heating
for 120 s. The vertical white dotted lines indicate the days of AR occurrence at the site, according to
Lapere et al. (2024). The horizontal dotted white line, black dashed line, and white dashed line indicate
the air-snow (AS), snow-ice (SI), and ice-water (IW) interfaces, respectively.






## 2.2. Observational and Reanalysis Datasets

Four other observational datasets are considered in this work: (i) satellite-derived SIE and sea-ice velocity (ii) daily true colour visible satellite images available at the National Aeronautics and Space Administration's (NASA's) WorldView website (Boller, 2024); (iii) ground-based observations at four weather stations located in the target region (Fig. 1b): at Mawson, Syowa, Mizuho and Relay stations; and (iv) sounding profiles at Syowa Station (Oolman, 2024).

SIE data are available at a resolution of 3.125 km and on a daily basis for the period June 2002 to present. It is estimated from the measurements collected by the Advanced Microwave Scanning Radiometer (AMSR) - Earth Observing Systems onboard NASA's Aqua satellite from June 2002 to October 2011, and from the observations taken by the AMSR2 onboard Japan Aerospace and Exploration Agency's Global Change Observation Mission - Water (GCOM-W; "Shizuko") satellite from July 2012 to present (Spreen et al., 2008). Sea-ice velocity vectors are available also daily at 62.5 km spatial resolution. This product is obtained from the measurements collected by the Special Sensor Microwave Imager/Sounder onboard the United States Air Force Defense Meteorological Satellite Program, the Advanced Scatterometer onboard the European Space Agency's Meteorological Operational Satellite, and the AMSR2 onboard the GCOM-W satellite, and is available from December 2009 to present (Lavergne et al., 2010). Warm and moist air intrusions impacting Antarctica can have substantial changes in SIE, with considerable sea-ice drift velocities that can exceed $50 \, \text{km day}^{-1}$ (e.g., Francis et al., 2021; Fonseca et al., 2023). Given this, both SIE and sea-ice velocity products are used to gain insight into the effects of the warm and moist air intrusions on the sea-ice state around the Mawson Station during the measurements.

Moderate Resolution Imaging Spectroradiometer (MODIS; Xiong et al., 2006; Gumley et al., 2010) true colour visible images are used to obtain additional high-resolution information on the SIE and its spatial variability (this is only possible in the absence of clouds, as otherwise the sea-ice and other features near sea level will not be visible). They also provided information on the presence of polynyas and the fine structure within the ice pack, as the spatial resolution is no lower than 1 km.

*In situ* observations at multiple Automatic Weather Stations (AWSs) are used in the analysis and model evaluation. These include: (i) 1-minute 2-m air temperature and humidity, 10-m horizontal wind velocity, and sea-level pressure (SLP) observations from the Mawson Station (67.6017ºS, 62.8753ºE); (ii) 1-minute measurements of meteorological parameters (2-m air temperature, SLP, 10-m horizontal wind velocity, and 2-m relative humidity) and radiation fluxes (surface upward and downward and shortwave and longwave) at the coastal Syowa Station (69.0053ºS, 39.5811ºE); and (iii) 10-minute SLP and horizontal wind velocity and 2-m air temperature and relative humidity observations at the inland Mizuho Station (70.70ºS, 44.29ºE) and Relay Station (74.017ºS, 43.062ºE). Also analyzed were data from atmospheric sounding profiles acquired twice daily (at 00 and 12 UTC) at Syowa Station.




In addition, the fifth generation of the European Centre for Medium Range Weather
Forecasting reanalysis (ERA-5) dataset (Hersbach et al., 2020) is used to investigate the large-
scale atmospheric circulation during the measurements and to analyze the surface energy budget
for the case study (11-16 November 2022). At a spatial resolution of 0.25º × 0.25º (~27 km) and
an hourly temporal resolution from 1940 to present, ERA-5 is regarded as one of the best reanalysis
products currently available over Antarctica and the Southern Ocean (Gossart et al., 2019; Dong
et al., 2020).

## 2.3. Numerical Models

Here we use version 4.3.3 of the Polar PWRF (Weather Research and Forecasting) model, a
version of the WRF model (Skamarock et al., 2019) optimized for the polar regions (Bromwich et
al., 2013; Hines et al., 2021; Xue et al., 2022; Zou et al., 2023), to simulate and investigate the AR
that impacted the Mac Robertson Land region on 14 November 2022. The model is run in a nested
configuration, with a 7.5 km horizontal resolution grid domain comprising Antarctica, the
Southern Atlantic Ocean, southern Africa and the southwestern Indian Ocean, and a 2.5 km
horizontal resolution grid domain extending from the Southern Ocean just south of South Africa
into coastal East Antarctica around the Mawson Station (Fig. 1a). The choice of resolution, in
particular the 2.5 km grid that covers the bulk of the AR and associated warm and moist air
intrusion into East Antarctica, reflects the findings of Box et al. (2023) and Francis et al. (2024).
These studies stressed the need to properly resolve the fine-scale structure of an AR due to the
possible presence of AR rapid-like features embedded in the convective region, which can generate
copious amounts of precipitation and hence have a substantial impact on the SMB of the ice. AR
rapids are narrow (5-15 km wide), elongated (100-200 km long) and shallow (~3 km deep) linear
features within the AR that propagate at high speed (>30 m s$^{-1}$) and last for more than 24 h. They
have been reported for an AR that impacted Greenland in September 2017 (Box et al., 2023) and
another that wreaked havoc in the Middle East in April 2023 (Francis et al., 2024). AR rapids are
distinct from mesoscale convective systems (MCSs; Houze, 2004; Feng et al., 2021; Nelli et al.,
2021), which propagate at a slower speed (10-20 m s$^{-1}$), typically do not last as long (6-10 h), and
generate broader (as opposed to linear) precipitation structures.

The physics schemes selected reflect the optimal model configuration for Antarctica and the
Southern Ocean (Zou et al. 2021a, 2021b, 2023): the two-moment Morrison-Milbrandt P3 cloud
microphysics scheme (Morrison and Milbrandt, 2015), with the Vignon adjustment to improve the
simulation of mid-level mixed-phase clouds over the Southern Ocean (Hines et al., 2021; Vignon
et al., 2021); the Mellor-Yamada-Nakanishi-Niino (MYNN) level 1.5 planetary boundary layer
(PBL) scheme (Nakanishi and Niino, 2006); the Rapid Radiative Transfer Model for Global
Circulation Models (RRTMG; Iacono et al., 2008) for shortwave and longwave radiation; the Noah
Land Surface Model (Chen and Dudhia, 2001; Tewari et al., 2004); the Kain-Fritsch cumulus
scheme (Kain, 1994) with subgrid-scale cloud feedbacks to radiation (Alapaty et al., 2012),





switched on in the 7.5 km grid only; and the Zeng and Beljaars (2005) surface skin temperature scheme. PWRF is run from 10 November 2022 at 00 UTC to 17 November 2022 at 00 UTC, comprising the strongest AR that impacted the site during July-November 2022, with the first day regarded as spin-up and the output discarded. The hourly outputs of the 7.5 km and 2.5 km grids are used for analysis. PWRF is driven by 6-hourly ERA-5 data, with the reanalysis' fractional SIE and ice concentration ingested into the model.

Due to the lack of availability of SIT in ERA-5, the model's default SIT value of 3 m is used in the PWRF simulations. The sea-ice albedo is parameterized as a function of air and skin temperature following Mills (2011), with the model explicitly predicting ST on sea ice. This control simulation ("PWRF") is repeated using as sea-ice concentration boundary conditions for the full 7.5 km and 2.5 km PWRF domains the 3.125 km-resolution daily product available at the University of Bremen website (UoB, 2024). For the SIT, and to contrast with the excessively thick 3 m default value, the range of values measured *in-situ* at the Khalifa SIMBA site on fast-ice off the Mawson Station towards the end of November, which is about 0.18 m to 0.30 m (Fig. 3a), is ingested into the model at all sea-ice covered grid-boxes. This simulation will be denoted as "PWRF_SIE_SIT" throughout the manuscript. Satellite-derived measurements suggest an overall similar range of values for the thickness of pack ice and fast-ice at multiple sites around Antarctica (Heil, 2006; Kacimi and Kwow, 2020; Li et al., 2022), justifying the usage of the same value for all sea-ice pixels in the model domain.

In order to prevent the large-scales in the model from drifting from the ERA-5 forcing fields, spectral nudging (Attada et al., 2021) is employed in both grids for spatial scales ≳ 1,000 km above ~800 hPa and excluding the boundary layer. Fields nudged include the horizontal wind components, the potential temperature perturbation, and the geopotential height. In the vertical, 60 levels are considered, with the lowest level above the surface at ~27 m and roughly 20 levels in the range of ~1-6 km. The higher resolution in the low- to mid-troposphere is crucial to correctly representing the fine-scale variability of the warm and moist air masses impacting the site, and associated cloud processes (Rauber et al., 2020; Finlon et al., 2020).

The moisture sources that contributed to the AR during 11-16 November 2022 are diagnosed based on 96-h back-trajectories obtained with the Hybrid Single-Particle Lagrangian Integrated Trajectory (HYSPLIT; Stein et al., 2015) model driven by ERA-5 reanalysis data.

## 2.4. Diagnostics and Metrics

The performance of the PWRF model is assessed with the verification diagnostics proposed by Koh et al. (2012) defined in Equations (1) to (5) below. These diagnostics are the (i) bias, $\underline{B}$, given by the mean discrepancy between the model forecasts, $F$, and the observations, $O$; (ii) normalized bias, $\mu$, defined as the ratio of the bias to the standard deviation of the discrepancy $B$ between $F$ and $O$ (following Koh et al. (2012), if $|\mu| < 0.5$, the bias makes a smaller contribution



to the Root Mean Square Error than the error variance and can therefore can be regarded as not
significant); (iii) correlation, $\rho$, which measures the phase agreement between the modelled and
observed data; (iv) variance similarity, $\eta$, an indication of the amplitude agreement between the
two signals; and (v) normalized error variance, $\alpha$, a diagnostic that combines phase and amplitude
errors. For a random forecast based on the climatological mean and variance $\alpha = 1$, the model
predictions can be deemed as practically useful if $\alpha < 1$. The $\rho$, $\eta$ and $\alpha$ skill scores are non-
dimensional, symmetrical with respect to observations and forecasts, and applicable to scalar and
vector fields - meaning that the model performance for scalars such as air temperature and vector
quantities such as the wind vector can be directly compared.  The verification diagnostics are:
$$B = F - O \quad (1)$$
$$\mu = \frac{<D>}{\sigma_D} \quad (2)$$
$$\rho = \frac{1}{\sigma_O \sigma_F} < (F - <F>) \cdot (O - <O>) >; \; -1 \leq \rho \leq 1 \quad (3)$$
$$\eta = \frac{\sigma_O \sigma_F}{\frac{1}{2}(\sigma_O^2 + \sigma_F^2)}; \; 0 \leq \eta \leq 1 \quad (4)$$
$$\alpha = 1 - \rho\eta = \frac{\sigma_D^2}{\sigma_O^2 + \sigma_F^2}; \; 0 \leq \alpha \leq 2 \quad (5)$$
ARs are identified based on the meridional Integrated Vapour Transport (vIVT; $\mathrm{kg\,m^{-1}\,s^{-1}}$),
which is the column integral of the water-vapour flux advected by the meridional wind. This
quantity is more appropriate for AR detection if the focus is on snowfall, which is the case here,
whereas for surface melting IVT is a better metric (Wille et al., 2019). It is quantified as:
$$vIVT = -\frac{1}{g} \int_{1000\,hPa}^{200\,hPa} qv \, dp \quad (6)$$
In equation (6), where $g$ is the gravitational acceleration ($9.80665\,\mathrm{m\,s^{-2}}$), $q$ is the specific humidity
($\mathrm{kg\,kg^{-1}}$), $v$ is the meridional wind speed ($\mathrm{m\,s^{-1}}$), and $dp$ is the pressure difference between adjacent
vertical levels (hPa). The AR outer boundaries are taken from Lapere et al. (2024), who used the
97[th] percentile of vIVT at a given grid-box and a minimum latitudinal extent of 20º to identify
ARs, from the Modern Era Retrospective Analysis for Research and Applications Version 2
dataset (MERRA-2; Gelaro et al., 2017). The ARs in that study were extracted globally for the
period 1980-2022, with the respective outlines made publicly available. The ARs for July-
November 2022 are considered in this work.



During the July to November 2022 study period, the Khalifa SIMBA site on fast-ice off the Mawson Station was affected by three ARs: on 14 July, 13 August and 14 November. The IVT and vIVT values around the Mawson Station, in particular the area-averaged values in a 2º × 2º domain centred around the station and obtained with MERRA-2 data to be consistent with the AR outlines, are highest for the 14 November AR. For this case, the maximum absolute IVT and vIVT values are 161 kg m$^{-1}$s$^{-1}$ and 112 kg m$^{-1}$s$^{-1}$, respectively, compared to 87 kg m$^{-1}$s$^{-1}$ and 39 kg m$^{-1}$s$^{-1}$ for the 13 August AR, and 148 kg m$^{-1}$s$^{-1}$ and 82 kg m$^{-1}$s$^{-1}$ for the 14 July AR. Based on these findings, the 14 November event is selected for more in-depth analysis and modeling in Section 4. Except for IVT and vIVT, for which MERRA-2 data are used as noted above, ERA-5 data are used to extract the other diagnostics outlined below.

For ARs to reach Antarctica, a large-scale circulation pattern that promotes the advection of warm and moist low-latitude air masses into the continent must be present. The leading mode of variability in the Southern Hemisphere extratropical atmospheric flow is the Southern Annular Mode (SAM; Marshall, 2003). This metric is based on the difference in mean sea-level pressure averaged over six stations at about 40ºS and six stations at about 65ºS, which are deemed representative of the zonal flow at the two latitudes. A positive index value indicates a stronger westerly flow in the Southern Hemisphere mid-latitudes, while a negative SAM phase is accompanied by an increase in blocking frequency (Oliveira et al., 2013). Atmospheric blocking promotes the development and propagation of ARs (Massom et al., 2004; Francis et al., 2021, 2022a; Wille et al., 2024). In this study, it is quantified using the blocking index ($BI$) proposed by Pook et al. (2013) and optimized over Antarctica by Wille et al. (2024c):

$$BI = 0.5\,(U_{35} + U_{40} + U_{65} + U_{70} - U_{50} - U_{60} - 2U_{55})\quad(7)$$

where $U_X$ is the geostrophic zonal wind computed from the 5-day running mean (in order to exclude temporary features) of the 500 hPa geopotential height at latitude $X$ºS. Mid-latitude blocking events correspond therefore to higher values of $BI$, with values in excess of 40 m s$^{-1}$ indicating a high degree of blocking.

The AR investigated in Section 4 originated over southern Africa, where tropical temperate troughs (TTTs), which arise from the interaction of mid-latitude baroclinic weather systems and tropical convection (Hart et al., 2013), are a regular occurrence. In order to assess whether a TTT event took place during the study period, we use the TTT index proposed by Ratna et al. (2023), which is based on Outgoing Longwave Radiation ($OLR$) and meridional wind speed as defined in equations (8a) and (8b), respectively:

$$OLR = \{[(OLR_{E1} + OLR_{E2})/2] \times 0.4 - [(OLR_{W1} + OLR_{W2})/2] \times 0.6\}\quad(8a)$$





In Equation (8a), E1 and E2 correspond to regions over Madagascar and southeastern Africa (E1: 37º-42ºE, 12º-17ºS; E2: 45º-50ºE, 23º-15ºS), with W1 and W2 located to the southwest of E1 and E2, the former over South Africa and the latter just offshore (W1: 22º-32ºE, 24º-18ºS; W2: 32º-42ºE, 36º-28ºS). In a TTT event, there are higher values of OLR ahead of the trough (E1 and E2) and lower values in the region where the trough is typically located (W1 and W2), with the placement of E1-E2 and W1-W2 reflecting the southeast-northwest orientation of the trough. The 0.4 and 0.6 factors in equation (8a) are indicative of the regional strength of the anomalies between the east and west regions, with the latter generally stronger than the former. The associated meridional wind index is defined as:

$$WIND = V_W - V_E \quad (8b)$$

The 850 hPa meridional wind speed is averaged over the western region (0º-15ºE, 38ºS-27ºS) to the southwest of South Africa, and the eastern region (34º-46ºE, 38º-27ºS) to the southeast of South Africa. If a trough is present, the associated clockwise circulation will lead to southerly winds to its west and northerly winds to its east, giving a positive value of the wind index. A TTT event requires the OLR and wind indices computed using the area-averaged anomalies to exceed their climatological standard deviations by 1.5 and 0.5, respectively.

Besides blocking and TTTs, the poleward transport of warm and moist low-latitude air is linked to the strength of the attendant cyclone, which is itself modulated by the presence of tropopause polar vortices (TPVs). As detailed in Wille et al. (2024c), TPVs are characterized by a minimum in potential temperature and a maximum in potential vorticity at the dynamic tropopause (PV = 2 $\times 10^{-6}$ m$^2$ K s$^{-1}$ kg$^{-1}$ = 2 PV Units = 2 PVU in the Northern Hemisphere and -2 PVU in the Southern Hemisphere). When co-located with increased low-level baroclinicity, they can trigger cyclogenesis, with a deeper low promoting an enhanced poleward propagation of the warm and moist low-latitude air mass. The TPVs are identified using the TPVTrack (v1.0) software described in Szapiro and Cavallo (2018), here driven by ERA-5 data.

The extratropical circulation can be modulated by tropical forcing, such as thermal (heating and cooling) anomalies (Hoskins and Karoly, 1981; Hoskins et al., 2012). In order to explore whether this occurs during the case study, the stationary wave activity flux that indicates the direction of anomalous stationary Rossby wave propagation, defined in Takaya and Nakamura (2001), is derived (and plotted) as:

$$W_X = \frac{p \cos(\phi)}{2|U|} \left\{ \frac{U}{a^2 \cos(\phi)^2} \left[ \left( \frac{\partial \psi'}{\partial \lambda} \right)^2 - \psi' \frac{\partial^2 \psi'}{\partial \lambda^2} \right] + \frac{V}{a^2 \cos(\phi)} \left[ \frac{\partial \psi'}{\partial \lambda} \frac{\partial \psi'}{\partial \phi} - \psi' \frac{\partial^2 \psi'}{\partial \lambda \partial \phi} \right] \right\} \quad (9a) \text{ and}$$

$$W_Y = \frac{p \cos(\phi)}{2|U|} \left\{ \frac{U}{a^2 \cos(\phi)} \left[ \frac{\partial \psi'}{\partial \lambda} \frac{\partial \psi'}{\partial \phi} - \psi' \frac{\partial^2 \psi'}{\partial \lambda \partial \phi} \right] + \frac{V}{a^2} \left[ \left( \frac{\partial \psi'}{\partial \phi} \right)^2 - \psi' \frac{\partial^2 \psi'}{\partial \phi^2} \right] \right\} \quad (9b)$$






where $p$ is the ratio of the pressure level at which the W-vector is computed and 1000 hPa, $\phi$ is the
latitude, $\lambda$ is the longitude, $U$ and $V$ are the zonal and meridional climatological wind speeds,
respectively, $|U|$ is the climatological mean wind speed, and $\psi'$ is the streamfunction anomaly.

Variability in the ST, and perhaps to a lesser extent the SIT, is directly related to by the surface
mass balance (SMB), which can be expressed as

$$SMB = P - Q_{sfc} - M - Q_{snow} - D \quad (10)$$

where $P$ is the precipitation rate (mostly snowfall; mm w.e. day$^{-1}$), $Q_{sfc}$ is the surface
evaporation/sublimation rate, $M$ is the surface melt and runoff rate, $Q_{snow}$ is the blowing snow
sublimation rate, and $D$ is the blowing snow divergence rate term. Blowing snow refers to
unconsolidated snow moved horizontally across the ice surface by winds above a certain threshold
speed (Massom et al., 2001). As detailed in Francis et al. (2023), the $P$ and $M$ terms are directly
extracted from ERA-5, for which the reanalysis values are in close agreement with satellite-derived
estimates over Antarctica, while the remaining three ($Q_{sfc}$, $Q_{snow}$, $D$) are calculated using
parameterization schemes. Positive values of SMB indicate an accumulation of snowfall at the
site, while negative values represent a reduction due to melting, sublimation or wind erosion
processes, or a combination of the three.
**3. Sea-Ice and Snow Thickness Variability**
In the bottom panels in Fig. 3a the derived values of ST and SIT from 8 July to 30 November
2022 at the Khalifa SIMBA site on fast-ice off the Mawson Station are plotted. The SIT exhibits
a gradual increase starting on 8 July, peaking at 1.14-1.16 m from 19-24 October, followed by a
steady decline to 0.06-0.10 m at the end of November. These values are comparable to those
estimated for this region and time of the year using satellite-derived products, which are typically
in the range 0.50-1.50 m (Kacimi and Kwok, 2020). The ST on top of the ice, on the other hand,
exhibits pronounced day-to-day variations as high as 0.08 m, peaking in mid-August to early
September, and with values not exceeding 0.10 m from mid-September to the end of November.
These values are also in the range of those derived from satellite altimeter data (Kacimi and Kwok,
504    2020).


In order to explore whether atmospheric forcing could have played a role in the observed
variability in SIT and ST, the local SMB is estimated around the Khalifa SIMBA site on fast-ice
off the Mawson Station using ERA-5 data. The SIT appears to be mostly driven by the ocean
forcing, and involving both ocean-driven fast ice deformation and thermodynamic growth (Heil et
al., 1996; Haas, 2017), and to a lesser extent the seasonal solar cycle, with the annual SIT decrease
that initiates in early November coinciding with the time when the air temperatures regularly climb





above 265 K (Fig. 2a). The marked drop in SIT of 0.6 m from 20 November to 25 November seen
in the bottom panel of Fig. 3a corresponds to a period when the surface and air temperature climbed
above freezing at the site (Fig. 2a). On the other hand, a comparison of the ST observations and
the sea-ice SMB estimated from ERA-5 (Equation 10) data reveals good correspondence between
the two. In particular, instances of positive SMB values (based on ERA5) are typically associated
with and followed by an increase in the measured ST at the site (e.g., in early July, mid-August,
early and mid-October and mid-November), while negative SMB values from ERA5 are
accompanied by a decrease in the observed ST (e.g., in late July-early August and in late
September-early October).
. Foehn winds are unlikely to play a dominant role in the sea-iceSMB off the Mawson Station,
even though the SIMBA site is exposed to katabatic winds flowing seaward off the interior plateau
(Dare and Budd, 2001). This is evidenced in Fig. 3a, which shows that the sea-ice SMB is largely
controlled by precipitation ($P$), while in Foehn wind events, surface sublimation ($Q_{sfc}$) is the
predominant term (Francis et al., 2023). For the case study discussed in Section 4 (11-16
November; Fig. 3b), there is a 0.06 m increase in ST from 14-15 November while the observed
SIT increases by 0.04 m from 0.74 m to 0.78 m at the same time, returning to the previous levels
(0.74 m) on 19 November. The results in Fig. 3b show a clear link between the observed
measurements and the reanalysis' SMB for 14 November AR. The increase in SIT, on the other
hand, may be explained by the freezing of (some of) the snow on top of the sea ice, as the surface
and air temperatures were below freezing, around 265 K (Fig. 2a), and/or by metamorphic
processes that can transform snow into ice (Sturm and Massom, 2017). The possibility that the
added snow would depress the sea-ice surface to below sea-level, with the resulting flooding of
the snow and subsequent freezing of the slush increasing SIT is unlikely. This is because the
required conditions, namely a snow:ice thickness ratio in excess of 1:3, and an ocean water that is
warm, with a temperature exceeding 268 K, and saline, with a bulk salinity higher than 5 psu (Sturm
and Massom, 2017), are not met during this period.
Figure 4a shows that a few blocking high events occurred around the site during the
measurements, in particular in late July-early August, late September-early October, and during
the month of October, when the ST was decreasing (Fig. 3a). Zoomed-in plots around the time of
each AR passage highlight the occurrence of blocking in particular in August (Fig. 4d), which
actually coincided with the passage of two consecutive ARs during 10-12 and 13-15 August (Fig.
4f). Wille et al. (2024c) and Maclennan et al. (2023) stressed that the occurrence of blocking can
lead to the development of an "AR family" (or multi-AR) event. The passage of the two ARs also
coincided with an increase in air temperature by more than 10 K in a couple of days (Fig. 4e),
which is also noted for July. It is explained by the counterclockwise flow around high-pressure
systems and subsequent poleward advection of warm and moist low-latitude air masses. The most
prominent such instance is around 150º-180ºE in late November 2022, when blocking around 180º





led to an air temperature increase of more than 15 K to above freezing levels at some locations (cf.
Figs. 4a-b).
In Fig. 2, the timings of AR passages at the site i.e., 14 July, 13 August and 14 November, are
highlighted by vertical dashed lines. In particular and in the July and August events, during the
polar night, there is a marked increase in air temperature of up to 18 K as the low latitude air mass
reached the SIMBA site; this is also seen in the ERA-5 Hovmoeller diagrams (Fig. 4e). In the 14
November event, the increase is substantially reduced (by up to 3 K) as the air temperature is
already much higher i.e., typically between 263-268 K. The ST increases by up to 0.06 m within
1-2 days of the AR event, returning to pre-AR levels in the following 2-4 days. The small
magnitude effect may arise from an increase due to snowfall during the passage of the AR and a
decrease before and after the event due to evaporation/sublimation in response to the drier and
windier conditions or snow removal by katabatic winds (Fig. 3a). Other processes, such as snow
metamorphism, by which snow changes to sea-ice (Sturm and Massom, 2017), can also play a
role. In fact, strong katabatic winds have been observed to blow the snow away as quickly as it
falls on nearshore fast ice near the Syowa Station, resulting in very low accumulation close to the
coast (Kawamura et al., 1995), and off the Mawson Station as well (Dare and Budd, 2001). The
SIT does not show a clear response to the passage of the ARs, except for the 14 November AR
where a 0.04 m increase may arise from snow-ice interactions as noted before (Sturm and Massom,
2017). It is important to note that a longer measurement period would be needed for a robust link
between ARs and their effects on ST and SIT to be established.
The results in Figure 4 stress the role of atmospheric dynamics in modulating the ST at the Khalifa
SIMBA site on fast-ice off the Mawson Station , with the SIT largely controlled by ocean dynamics
(ocean-driven fast-ice deformation and thermodynamic growth) and seasonal variability in
incoming solar radiation.

(a)





**Figure 3: Surface Mass Balance and SIMBA Observations:** (a) Surface mass balance (mm w.e. hr$^{-1}$) from ERA-5 (top two plots) averaged over 66.5º-68.5ºS and 62.5º-63.5ºE and ST and sea-ice thickness



(SIT; m) from the SIMBA measurements (bottom two plots) for the period 8 July to 30 November 2022.
(b) is as (a) but for 10-20 November 2022. The local SMB terms plotted are the SMB, precipitation (P),
snowmelt (M), surface sublimation ($Q_{sfc}$), blowing snow sublimation ($Q_{snow}$), and blowing snow
divergence (D).


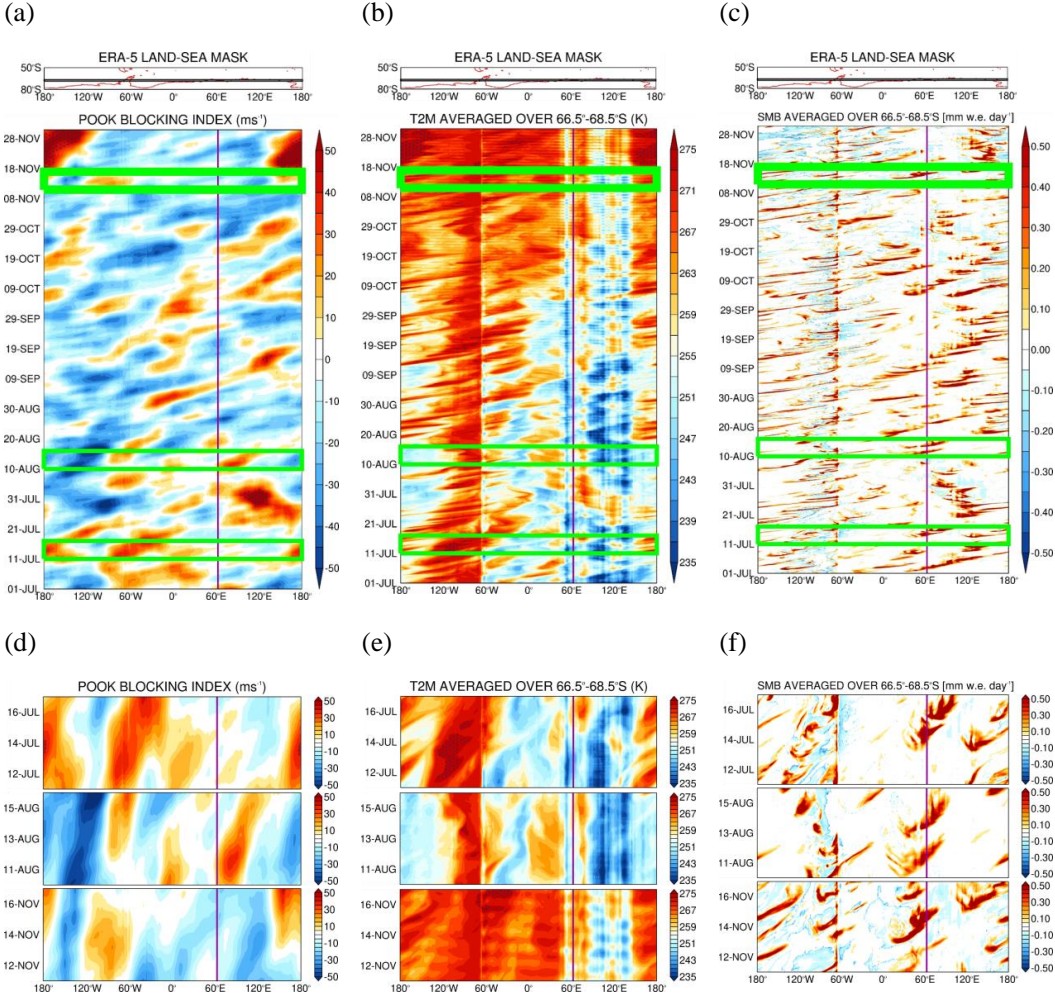

**Figure 4: Atmospheric dynamics and thermodynamics during the Observational Period:** (a) Pook
blocking index ($m\,s^{-1}$) for July-November 2022. The vertical purple line gives the approximate longitude
of the measuring site. Regions where the index exceeds $40\,m\,s^{-1}$, an indication of a high degree of
blocking, are stippled. The green rectangles indicate the periods when an AR impacted the site: 11-16
July, 10-15 August, and 11-16 November. The latter is considered for modeling and is highlighted with
a thick line. Above the Hovmoeller plot, the land-sea mask as seen by ERA-5 is plotted in red and the
averaging region is highlighted with a black rectangle. (b) is as (a) but for air temperature (K) averaged
over 68.5°-66.5°S. The sharp transition in the temperature field around 60°W arises due to the presence
of the Antarctic Peninsula (landmass). The stipple indicates regions and times when the temperature is



above freezing (273.15 K). (c) is as (b) but for the SMB defined in equation (10). (d)-(f) are as (a)-(c) but zooming in for each of the three periods.


## 4. Case Study: 11-16 November 2022

The strongest AR to impact the site during July-November 2022 occurred on 14 November. In
Section 4.1, the large- and regional-scale environment that promoted the development of the AR
is investigated, while in Section 4.2 the results of the PWRF simulations are discussed.

### 4.1 Large-Scale Atmospheric Patterns

The period 10-19 November 2022 is characterized by a strong wavenumber 3 pattern in the
Southern Hemisphere mid-latitudes (Fig. 5a), in association with a positive SAM phase. In fact,
the SAM index for November 2022 is the third highest since 1979, and is more than 1.5 standard
deviations above the 1979-2021 climatological mean (Fig. S1a). The stationary wave activity flux
vectors in Fig. 5a show little wave propagation from the tropics into the Southern Hemisphere
mid-latitudes, with a prevailing zonal propagation within the wavenumber #3 pattern. This is also
evidenced by the strong westerly flow around Antarctica (Figs. 5c-d). One of the reasons for the
positive SAM is the La Niña that was taking place at the time, the third consecutive La Niña year
after the 2018-2019 El Niño (NOAA/NWS, 2024). La Niña events favour a stronger than normal
Amundsen-Sea Low (Raphael et al., 2016), as was the case during November 2022 (Fig. 5b). In
the previous month (October) it was even deeper, with a cyclone in the South Pacific Ocean
reaching a sea-level pressure of 900 hPa, making it the strongest extratropical cyclone since the
start of the satellite era in 1980 to date (Lin et al., 2023).

(a)                                       (b)



**Figure 5: Large-Scale Circulation during 10-19 November 2022:** (a) 200 hPa stream-function anomalies (shading; $10^6$ m$^2$s$^{-1}$), with respect to the hourly 1979-2021 climatology, and the stationary W vectors (Takaya and Nakamura, 2001; arrows; m$^2$s$^{-2}$) averaged over 10-19 November 2022. (b) Sea-level pressure (shading; hPa) and 10-m wind vectors (arrows; m s$^{-1}$) anomalies for the same period. (c) and (d) show the 200 hPa and 850 hPa wind speed (shading; m s$^{-1}$) and vectors (arrows) averaged over the same period. The star gives the location of the Mawson Station (67.5912°S, 62.8563°E).







North of Mawson Station, a pressure dipole is present around 40º-65ºS (Figs. 5-b), with a ridge to
the east and a trough to the west, with both features more than two standard deviations away from
the climatological mean (Fig. 6e). The interaction between the subtropical jet and polar jet (Fig.
5c) led to the development of a jet streak, a localized maximum in the strength of the flow, on 13-
14 November that promoted an intensification of the low. Despite its slow eastward movement
and anomalous strength, the meridional extent of the ridge from East Antarctica to southeastern
Madagascar may explain why it is not detected by the Pook blocking index, Fig. 4a and Equation
(7), as the westerly flow at 35ºN and 40ºN is also weaker. In any case, this pressure dipole fosters
the transport of warm and moist low-latitude air across the SIMBA site and is conducive to the
development of ARs (Francis et al., 2022b; Gorodetskaya et al., 2023). The one that developed on
14 November 2022 is particularly remarkable, extending from tropical Africa into the Southern
Ocean and East Antarctica (Figs. 6a-b). The IVT anomalies at 06 UTC on 14 November exceed 50
$kg\,m^{-1}\,s^{-1}$ around the SIMBA site and $400\,kg\,m^{-1}\,s^{-1}$ further north along the AR (Fig. 6b), with the
hourly IVT on this day being in the top 1% of the climatological distribution (Fig. 6b), an
attestation to the extreme nature of this event. The air temperature anomalies are also noteworthy,
exceeding 8 K in parts of East Antarctica just west of the site (Fig. 6d), where they are more than
two standard deviations above the 1979-2021 climatological mean (not shown).
This AR and associated warm and moist air intrusion left a considerable imprint on the weather
conditions over East Antarctica around and to the west of the Mawson Station. Furthermore, it had
an important effect on the sea ice in the region. As seen in Figs. S3a-b, there was a considerable
reduction in SIE from 14 to 17 November both around coastal Antarctica and upstream. The sea-
ice vectors in Figs. S3c-d show an equatorward movement north of Mawson Station from 11-13
November (prior to the event) and southward movement from 14-16 November (post event) at
speeds in excess of $25\,km\,day^{-1}$, the latter an order of magnitude larger than that estimated during
12-14 November at the same site These sea-ice drift velocities are comparable to those observed
in the western Ross Sea in late April 2017 (Fonseca et al., 2023), and are associated with the
changing wind field in response to the shift in the position of the mid-latitude weather systems in
the region (Fig. 7).
The southeast-northwest convective band over southern Africa is a potential TTT event, resulting
from the interaction of mid-latitude weather systems with tropical convection. Such TTTs are
known to precondition the environment for the development of ARs, as in the March 2022 East
Antarctica "heat" wave (Wille et al. 2024a,b). In order to quantify its strength and check whether
a TTT event took place during the study period, the TTT index put forward by Ratna et al. (2023),
which is based on OLR and meridional wind (equations 8a,b), is utilized (Fig. S1b). While the
meridional wind index does exceed half of its climatological standard deviation during 12-13
November, the OLR index does not meet its condition of being higher than 1.5 the climatological
standard deviation. Hence, no TTT event occurred during 10-20 November 2022. Having said this,
tropical and subtropical moisture contributed to the warm and moist air intrusion that impacted





East Antarctica. This is evident in the back-trajectories obtained with HYSPLIT forced with ERA-
5 data (Fig. S2). While at lower levels (500 m and 1500 m) the moisture came from the Southern
Ocean, at 2500 m it originated in the subtropics just south of South Africa before rising just north
of the Mawson Station when this moist air mass encountered the colder and drier katabatic airflow.
Even at 500 m, the dry air parcels descending the Antarctic plateau into the Southern Ocean are
moistened over the water before turning back to Antarctica and reaching the site (Figs. S2b-e).
Several studies report on ARs impacting Antarctica being fed by subtropical moisture, such as the
February 2011 (Terpstra et al., 2021) and the November-December 2018 (Gorodetskaya et al.,
2020) ARs over East Antarctica, and the February 2022 AR over the Antarctica Peninsula
(Gorodetskaya et al., 2023).

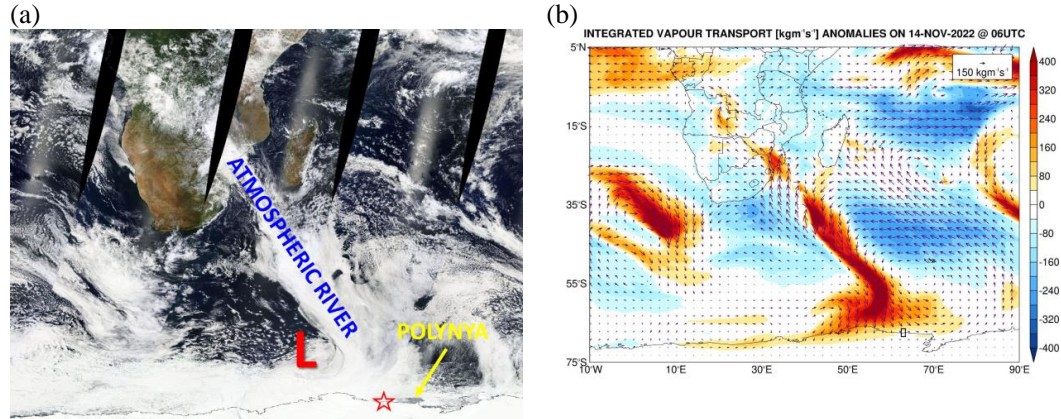

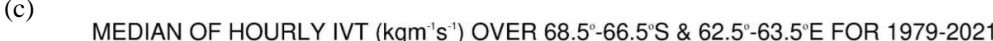

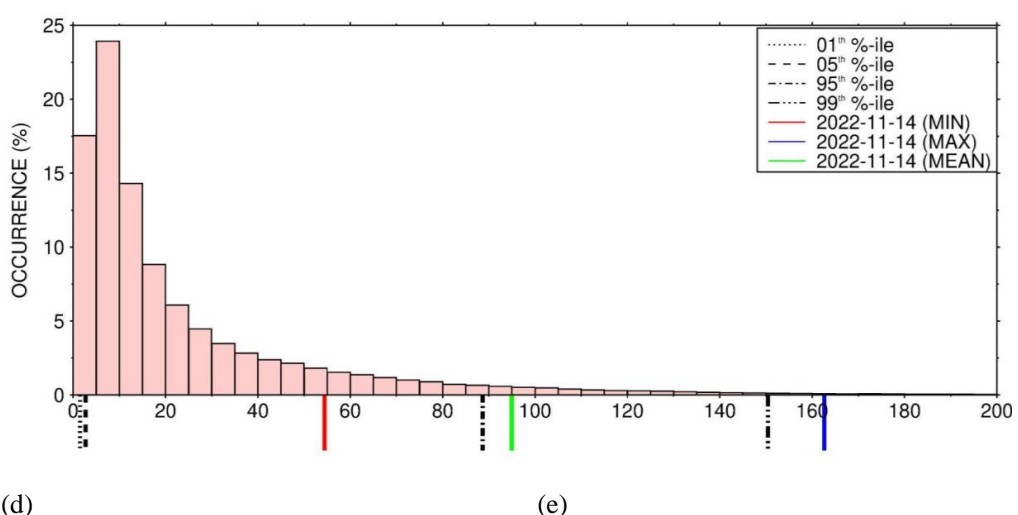

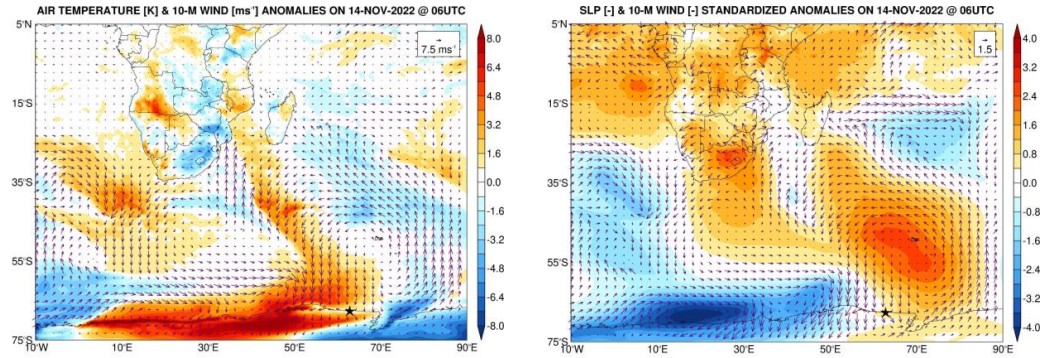

**Figure 6: Atmospheric River on 14 November 2022:** (a) MODIS visible image on 14 November 2022 over the domain 10ºW-90ºE and 5ºN-75ºS. The location of the Atmospheric River, Mawson Station (star) and a coastal polynya to the east of the station are highlighted. Image Credits: NASA WorldView. (b) Integrated Vapour Pressure (IVT; kg m$^{-1}$ s$^{-1}$) anomalies, the shading gives the magnitude and the arrows the vectors, on 14 November 2022 at 06 UTC with respect to the hourly 1979-2021 climatology from ERA-5. (c) Histogram of the median hourly IVT for the domain 68.5º-66.5ºS and 62.5º-63.5ºE, black box in (b), for 1979-2021. The dotted, dashed, dotted-dashed and dashed-dotted-dotted lines give the 1$^{st}$, 5$^{th}$, 95$^{th}$ and 99$^{th}$ percentiles, respectively, while the red, green and blue lines indicate the minimum, mean and maximum hourly values on 14 November 2022. (d) is as (b) but for the air temperature (shading; K) and 10-m wind vectors (arrows; m s$^{-1}$), while in (e) the shading gives the sea-level pressure and the arrows give the 10-m wind vector standardized anomalies.


653       Figures 5-6 provide a summary of the weather conditions during 10-20 November 2022, with
Figure 6 focusing on the AR event that peaked on 14 November. In order to gain insight into this
AR event, it is important to assess the temporal evolution of the atmospheric circulation prior to
and during the event itself. This is achieved in Figure 7, which shows multiple fields every 12 h
from 13 November at 18 UTC to 15 November at 06 UTC. At 18 UTC on 13 November (Fig. 7a),
a low-pressure system is centered west of the site, coincident with a TPV (highlighted in the figure)
which came from the Antarctic plateau (full track shown in Fig. S1c), and a ridge to its east. The
TPV helps the surface low to intensify, together with the jet streak at upper levels (Fig. 5c). The
pressure dipole promotes the southward advection of a warmer and moist low-latitude air mass
into the Southern Ocean, as noted by the hatching that highlights regions where the IVT exceeds
250 kg m$^{-1}$ s$^{-1}$. A secondary low, which develops early on 14 November (highlighted in Fig. 7b also
also noted by the additional sea-level pressure contour) is not co-located with a TPV. Instead, it is
driven by the interaction of the warm and moist air mass from the west and northwest around the
low pressure with that from the northeast around the ridge - and also closer to the Antarctic coast
with the drier and colder katabatic flow blowing from the continent. The maximum Eady growth
rate, a measure of baroclinicity (Hoskins and Valdes, 1990), at 850 hPa exceeded 3 day$^{-1}$ on 14
November (not shown), indicating a highly baroclinic environment.

671       Figures 7b-c show cyclonic Rossby wave breaking, with the secondary low exhibiting little
eastward movement owing to the presence of a strong ridge to the east (Fig. 6e) and instead shifting





southwards towards Antarctica. The incursion of the higher low-latitude potential temperature
values into East Antarctica (Figs. 7b-d) is consistent with the warmer (Fig. 6d) and moister (Figs.
6b-c) conditions in the region. The flow became westerly and the warm and moist air intrusion
weakened and shifted eastwards from 14 to 15 November (Figs. 7c-d), with another warm and
moist air intrusion (albeit weaker) developing to the northwest of the site (Fig. 7d) later impacting
the area on 16-17 November (not shown). Fig. 7 shows more than one episode of intrusion of low-
latitude air masses into Antarctica. For example, on 14-16 November a warm and moist air
intrusion reached Victoria Land just to the west of the Ross Sea (Figs. 7c-d). Such occurrences are
more common in an amplified pattern, and can be aided by TPVs that act to strengthen the
attendant cyclone (Wille et al., 2024c).

(a)                                                    (b)

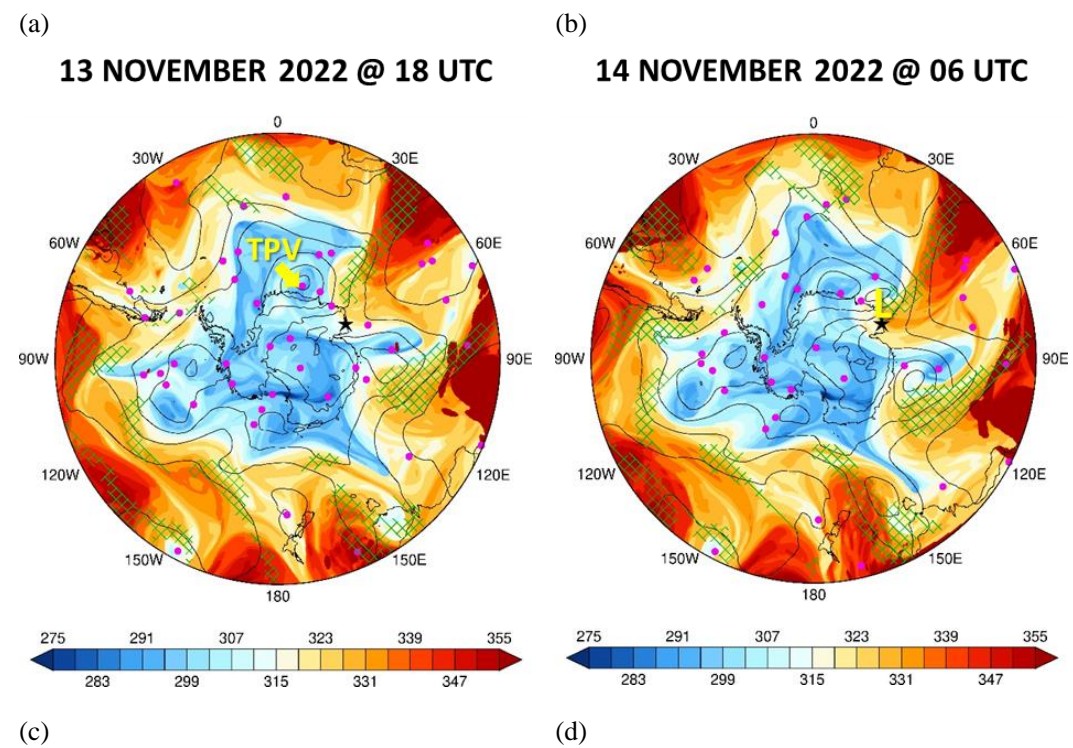

(c)                                                    (d)



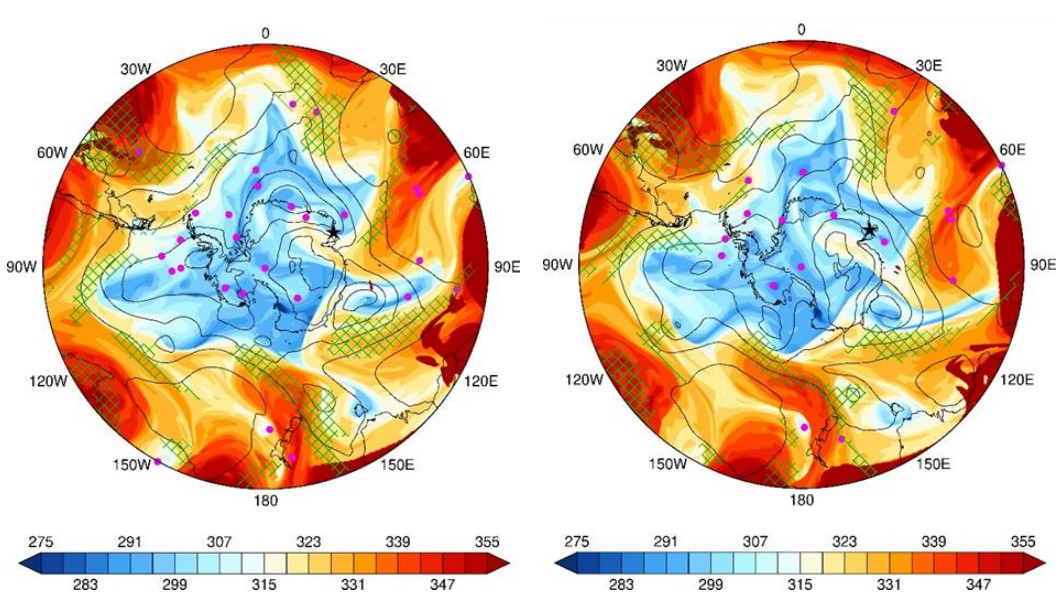

**Figure 7: Evolution of Atmospheric State during 13-15 November 2022:** Potential temperature ($\theta$; shading; K) on the dynamical tropopause (PV = -2 PVU), sea-level pressure (black contours; every 15 hPa starting at 900 hPa) and integrated vapour transport (IVT; hatching if > 250 kg m$^{-1}$ s$^{-1}$) on (a) 13 November at 18 UTC, 14 November at (b) 06 UTC and (c) 18 UTC, and (d) 15 November at 06 UTC. The purple dots indicate the location of tropopause polar vortices (TPV) at the respective times. The TPV and secondary low discussed in the text are highlighted in panels (a) and (b), respectively.


## 4.2 PolarWRF Simulation

In this subsection, the focus is on the modeling experiments. In Section 4.2.1, the PWRF predictions are evaluated against in-situ measurements at four stations in East Antarctica given in Fig. 1b, while in Section 4.2.2 the emphasis is on the additional insight the higher-resolution model data gives on the mid-November 2022 AR event.

### 4.2.1 Evaluation of PolarWRF

The PWRF simulations for 11-16 November 2022 are evaluated against in-situ meteorological observations at the Mawson, Syowa, Mizuho and Relay stations, in addition to surface radiation fields at Syowa Station. Fig. 8 shows the time-series of hourly data for the Mawson and Syowa stations, with the corresponding time series for the other two stations given in Fig. S4. A quantitative assessment of the model performance for all stations and variables is presented in Table 1.




The PWRF simulates the weather conditions well at Mawson (Fig. 8a), Syowa (Fig. 8b), Mizuho
(Fig. S4a) and Relay (Fig. S4b) stations for 11-16 November 2022. In particular, (i) the observed
variability in sea-level pressure is well replicated, with the model correctly capturing the time of
passage and strength of the secondary cyclone on 14 November (Figs. 7b-c) at all sites; (ii) the
warmer, more moist and windier conditions on 13-15 November are predicted by the model at all
sites; and (iii) it captures the reduction in the surface downward shortwave radiation flux by about
$200\,\mathrm{W\,m^{-2}}$, or a third of its value, and the increase in the downward long-wave radiation flux by
up to $90\,\mathrm{W\,m^{-2}}$ at Syowa in association with the warm and moist air intrusion. An inspection of
Table 1 reveals that, by and large, the normalized bias $\mu$ is smaller than 0.5, indicating the (small
magnitude) biases can be regarded as not significant, while the normalized error variance $\alpha$ does
not exceed 1 for all fields and stations (except for the wind vector at the higher-elevation Relay
Station), indicating that the PWRF predictions can be regarded as trustful. The performance of
PWRF for this site and event is comparable to that for the McMurdo Station in early January 2016
(Hines et al., 2019), for West Antarctica in early to mid-January 2019 (Bromwich et al., 2022),
and for the Antarctic Peninsula for May-June 2019 and January 2020 (Matejka et al., 2021). This
is a reflection of the improvements made to PWRF by the model developers, with the aim of
optimizing its performance and skill over Antarctica (e.g., Hines et al., 2021).

A closer inspection of Figs. 8 and S5 reveals some discrepancies in the PWRF predictions. For
example, at Syowa Station, the model has a tendency to over-predict the air temperature by ~1-2
K. While the downward shortwave radiation flux is generally well captured by the model, the
upward shortwave flux has a significant negative bias of ~$68\,\mathrm{W\,m^{-2}}$, which can arise e.g. from an
underestimation of the observed surface albedo by around 10% (roughly 0.84 for observations and
0.75 for PWRF for 11-16 November). This suggests the need to properly represent land surface
properties in the model, which has been highlighted by other studies (e.g., Hines et al., 2019) The
lower albedo in PWRF leads to a positive bias in the net shortwave radiation flux, which is
consistent with the warmer air temperatures and the enhanced upward longwave radiation flux
biases of ~$11\,\mathrm{W\,m^{-2}}$. At all four stations, the predicted wind direction is shifted clockwise by 45º-
90º compared to that observed, with this mismatch being more evident at Relay Station located on
the Antarctic plateau more than $3,000\,\mathrm{m}$ above sea-level (Fig. 2b). This can be attributed to an
incorrect representation of the surface topography which, as for surface properties such as the
albedo, exhibits a complex spatial heterogeneity in the region (Lea et al., 2024). Despite these
issues, both the magnitude and variability of the observed wind speed are generally well
represented by PWRF (Figs. 8 and S3). The more offshore wind direction at the coastal Mawson
and Syowa stations reflect a stronger katabatic wind regime that acts to slow the poleward
movement of the warm and moist low-latitude air mass, which is consistent with the dry bias of
$0.11\text{-}0.16\,\mathrm{g\,kg^{-1}}$. In fact, and in particular at the Mawson Station, when the model overpredicts the
strength of the near-surface wind (e.g., around $00\,\mathrm{UTC}$ on 12 and 16 November and between 18-



24 UTC on 13 November) from an offshore direction, there is a cold and dry bias, confirming the
occurrence of an enhanced katabatic airflow.
Table 1 also reveals that the control and model simulation with updated SIE and SIT yield
similar skill scores, a fact that is confirmed by the time-series in Figs. 8 and S4. This suggests that
a more realistic representation of the sea-ice state, and at least for this particular event and model
configuration, does not translate into more accurate predictions. By and large, the results in Figs.
8 and S4 indicate a tendency for drier and windier conditions compared to observations. This has
been reported in a number of PWRF studies (e.g., Wille et al. 2016, 2017; Vignon et al., 2019),
and has been attributed to too much boundary layer mixing in the model. An optimized PBL
scheme, which at least partially corrects the excessive mixing, and/or a more sophisticated land
surface model that more accurately represents the boundary layer and surface processes have to be
considered to address the aforementioned biases.
Besides ground-based observations, sounding data are available at Syowa Station every 12 h
(Fig. S5a) and can be compared with the hourly PWRF predictions (Figs. S5b-c). The model
captures the timing of the arrival of the warm and moist air mass on 14 November well, as
evidenced by the higher values of $\theta_E$ (280-290 K) and relative humidity (90-100%). The
northwesterly flow between 750 and 950 hPa late on 14 November is also simulated by PWRF,
even though the wind direction in the model tends to be more from an easterly component
compared to observations. The results in Figs. 8 and S4-S5 and Table 1 reveal a good PWRF
performance in the study area for the period 11-16 November 2022. In the next subsection the
model simulations are used to gain further insight into the dynamics for this event. The simulation
with the updated SIE and SIT was used for this purpose.

    (a)                                              (b)



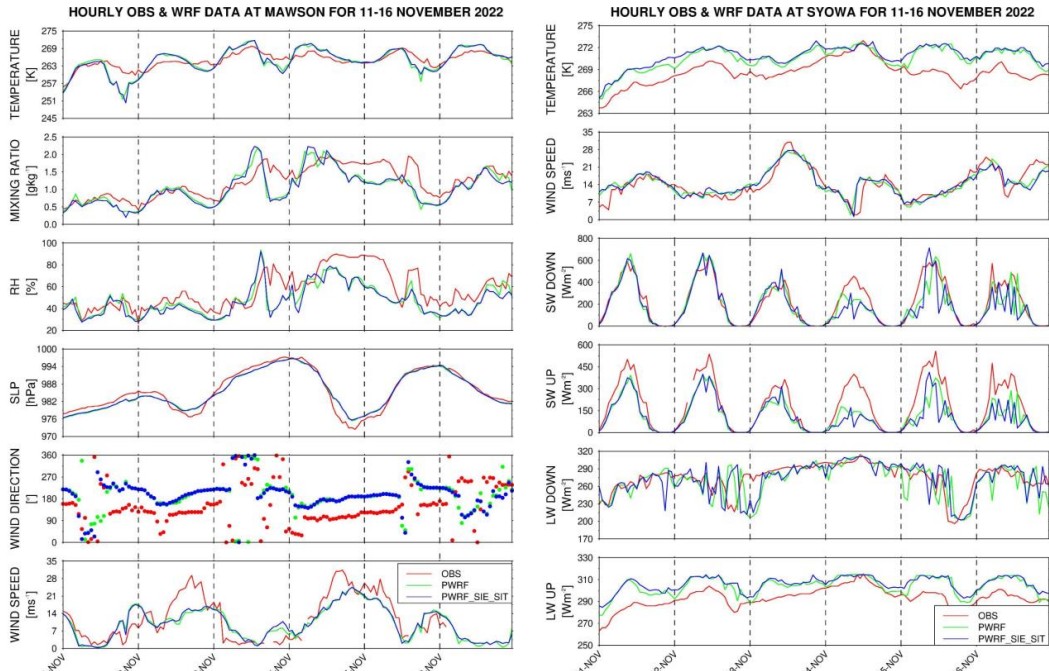

**Figure 8: Evaluation of PolarWRF against ground-based observations:** (a) Hourly air temperature (ºC), water vapour mixing ratio (g kg$^{-1}$), relative humidity (RH; %), sea-level pressure (SLP; hPa) and horizontal wind direction (º) and speed (m s$^{-1}$) from observations (red) and for the control (green) PWRF simulation and the one with updated SIE and SIT (blue) for 11-16 November 2022 at the Mawson Station. (b) is as (a) but for the hourly air temperature (K), horizontal wind speed (m s$^{-1}$), and surface downward and upward shortwave and longwave radiation fluxes (W m$^{-2}$) at the Syowa Station. The location of the stations is given in Fig. 1b.



| Variable | Station | Bias | $\mu$ | $\rho$ | $\eta$ | $\alpha$ |
|---|---|---|---|---|---|---|
| **Air Temperature** | Mawson | -0.04 K (-0.10 K) | -0.01 (-0.04) | 0.81 (0.82) | 0.90 (0.88) | 0.27 (0.27) |
| | Syowa | 1.89 K (2.26 K) | 1.77 (2.16) | 0.77 (0.77) | ~1.0 (0.99) | 0.24 (0.24) |
| | Mizuho | -0.44 K (-0.26 K) | -0.22 (-0.14) | 0.95 (0.95) | 0.98 (0.98) | 0.07 (0.07) |
| | Relay | 1.13 K (0.96 K) | 0.32 (0.26) | 0.81 (0.80) | 0.99 (~1.0) | 0.19 (0.20) |
| **Water Vapour** | Mawson | -0.16 g kg$^{-1}$ | -0.52 | 0.77 | ~1.0 | 0.24 |




| **Mixing Ratio** | | (-0.18 g kg⁻¹) | (-0.56) | (0.76) | (~1.0) | (0.24) |
|---|---|---|---|---|---|---|
| | Syowa | -0.11 g kg⁻¹ (-0.04 g kg⁻¹) | -0.34 (-0.13) | 0.83 (0.81) | 0.98 (0.98) | 0.19 (0.21) |
| | Mizuho | - (-) | - (-) | - (-) | - (-) | - (-) |
| | Relay | 0.02 g kg⁻¹ (0.02 g kg⁻¹) | 0.28 (0.24) | 0.73 (0.72) | 0.99 (0.98) | 0.28 (0.29) |
| **Wind Vector** (Bias and $\mu$ are for wind speed) | Mawson | -1.24 m s⁻¹ (-1.17 m s⁻¹) | -0.23 (-0.22) | 0.35 (0.34) | 0.97 (0.96) | 0.66 (0.67) |
| | Syowa | 0.13 m s⁻¹ (0.15 m s⁻¹) | 0.04 (0.04) | 0.62 (0.59) | 0.99 (0.99) | 0.39 (0.41) |
| | Mizuho | 1.39 m s⁻¹ (1.23 m s⁻¹) | 0.83 (0.69) | 0.60 (0.60) | 0.98 (0.97) | 0.41 (0.42) |
| | Relay | 0.41 m s⁻¹ (0.46 m s⁻¹) | 0.25 (0.28) | -0.73 (-0.71) | 0.98 (0.98) | 1.72 (1.70) |
| **Surface Pressure** | Mawson | -4.22 hPa (-4.20 hPa) | -2.75 (-2.81) | 0.98 (0.98) | ~1.0 (~1.0) | 0.03 (0.03) |
| | Syowa | 4.03 hPa (3.89 hPa) | 2.75 (2.62) | 0.99 (0.99) | ~1.0 (~1.0) | 0.02 (0.02) |
| | Mizuho | -0.67 hPa (-0.69 hPa) | -0.82 (-0.83) | 0.99 (0.99) | ~1.0 (~1.0) | 0.01 (0.01) |
| | Relay | 2.24 hPa (2.23 hPa) | 3.20 (3.19) | 0.99 (0.99) | ~1.0 (~1.0) | 0.01 (0.01) |
| **Downward SW** | | -24.89 W m⁻² (-36.47 W m⁻²) | -0.30 (-0.37) | 0.90 (0.86) | ~1.0 (~1.0) | 0.10 0.14 |
| **Upward SW** | | -68.43 W m⁻² (-74.77 W m⁻²) | -0.86 (-0.83) | 0.90 (0.86) | 0.93 (0.92) | 0.17 (0.21) |
| **Downward LW** | Syowa | -4.40 W m⁻² (-2.00 W m⁻²) | -0.19 (-0.09) | 0.63 (0.63) | 0.98 (0.99) | 0.38 (0.38) |
| **Upward LW** | | 10.71 W m⁻² (12.73 W m⁻²) | 1.69 (2.17) | 0.73 (0.75) | ~1.0 (0.97) | 0.27 (0.27) |

**Table 1: Verification diagnostics with respect to station data**: Bias, normalized bias ($\mu$), correlation



($\rho$), variance similarity ($\eta$) and normalized error variance ($\alpha$) for air temperature, water vapour mixing
ratio, horizontal wind vector and sea-level pressure for the Mawson, Syowa, Mizuho and Relay stations
for 11-16 November 2022. For the Syowa Station, the scores are also given for the surface downward and
upward shortwave and longwave radiation fluxes. Humidity measurements are not available at the
Mizuho Station for this period. The first value gives the score for the control simulation, while the one in
parenthesis is for the simulation with updated SIE and SIT. The model values are those at the closest
model grid-point to the location of the station, and the evaluation is performed for hourly data. The
correspondent time-series are given in Figs. 7 and S3.

**4.2.2 Insights into the Dynamics and Effects of the AR**
One of the motivations for the high-resolution (2.5 km) innermost grid is to check for the
presence of AR rapids (Box et al., 2023; Francis et al., 2024). Figs. 9a-c show a hovmoeller plot
of the vertical velocity at 700 hPa, the 850 hPa equivalent potential temperature, and precipitation
rate averaged over 40º-50ºE, a latitude band that comprises the bulk of the AR (Figs. 7a-b and
10a). No AR rapids are seen in all fields as well as in the vertical profiles (Fig. S5b). Instead and
from 12 UTC on 13 November to 12 UTC on 14 November, the AR exhibits mesoscale frontal
wave structures between 50º-60ºS, with an increase in precipitation just off the Antarctica coast at
~65º-67.5ºS, Fig. 9c, likely arising from the interaction of the low-latitude air mass with the
katabatic wind flow. At about 50ºS at 18 UTC on 13 November, there are two propagating
atmospheric structures: one moving southwards and reaching Antarctica on 14 November, and
another moving northwards, reaching 40ºS at about the same time (Figs. 9a-c). The initial AR band
breaks into two pieces, with one moving southwards into Antarctica, the one discussed here, while
the counter-clockwise circulation associated with a ridge moving in from the west slows down and
gradually pushes the northern part equatorwards (cf. Figs. 10a and 10c). A similar contrasting
poleward and equatorward propagation is seen on 15-16 November at about 65ºS, here driven by
the interaction of the katabatic winds off Antarctica with the flow around the ridge to the east (Figs.
5b and 6e).
On top of surface evaporation from the subtropics (Fig. S2), the convergence of the flow
around the low-pressure system to the west and the ridge to the east helped feed the AR and
associated warm and moist air mass (Fig. 7). This can be seen in Figs. 10a-b. The zonal moisture
transport in Fig. 10b highlights the convergence of the westerly flow at 10-15 m s$^{-1}$ associated with
equivalent potential temperature ($\theta_E$) values of 280-285 K, and the more moist easterly flow around
the high with zonal wind speeds in excess of 25 m s$^{-1}$ and $\theta_E$ ~ 290-300 K, as this air mass comes
directly from the tropics. Precipitation rates in excess of 3 mm hr$^{-1}$ are simulated by the model at
12 UTC on 13 November along the AR (Fig. 10a). As the moisture plume moved closer to the
Antarctic coast, it interacted with the katabatic wind regime. This is evident in Fig. 10d, with the
drier ($\theta_E$ ~ 275-280 K) and strong (meridional wind speeds in excess of 40 m s$^{-1}$) flow from
Antarctica converging with the slower (20-30 m s$^{-1}$) and more moist ($\theta_E$ ~ 280-290 K) flow from
lower-latitudes. This convergence led to precipitation rates in excess of 3 mm hr$^{-1}$ just north of the
Mawson Station (Fig. 10c).



The pattern of the precipitation field, which has a gap-core structure, reflects the complex
topography of the region (Fig. 1b). The evolution of the interaction between the warm and moist
southward-moving and the colder and drier northward-moving air masses is displayed in Figs. 9d-
f, where the meridional wind speed, $\theta_E$ and precipitation rate are averaged over 55º-65ºE; the band
of strong convergence (Fig. 10c). On 12 November, and in particular on 14-15 November, the
strong southerly winds with speeds in excess of $20\,\mathrm{m\,s^{-1}}$ converged with, at times, an equally strong
northerly flow, Fig. 9d, with precipitation around the convergence line, Fig. 9f, where $\theta_E$ values
exhibit steep meridional gradients that can exceed $25\,\mathrm{K}$, Fig. 9e. The katabatic winds on 12 and
14-15 November led to the opening up of a polynya east of the site (Fig. 6a). Coastal polynyas are
a regular and persistent feature at certain locations around Antarctica owing to the steep coastal
terrain and topographic channeling of katabatic winds (Barber and Massom, 2007), with warm and
moist air intrusions also playing a role in their spatial extent (Fonseca et al., 2023).
The results in Figs. 9d and 10c-d suggest that it can be difficult for ARs and associated warm and
moist air intrusions to reach this region of East Antarctica owing to the interaction with the strong
katabatic flow. This has been highlighted for other regions of East Antarctica (e.g., Terpstra et al.,
2021; Gehring et al., 2022).

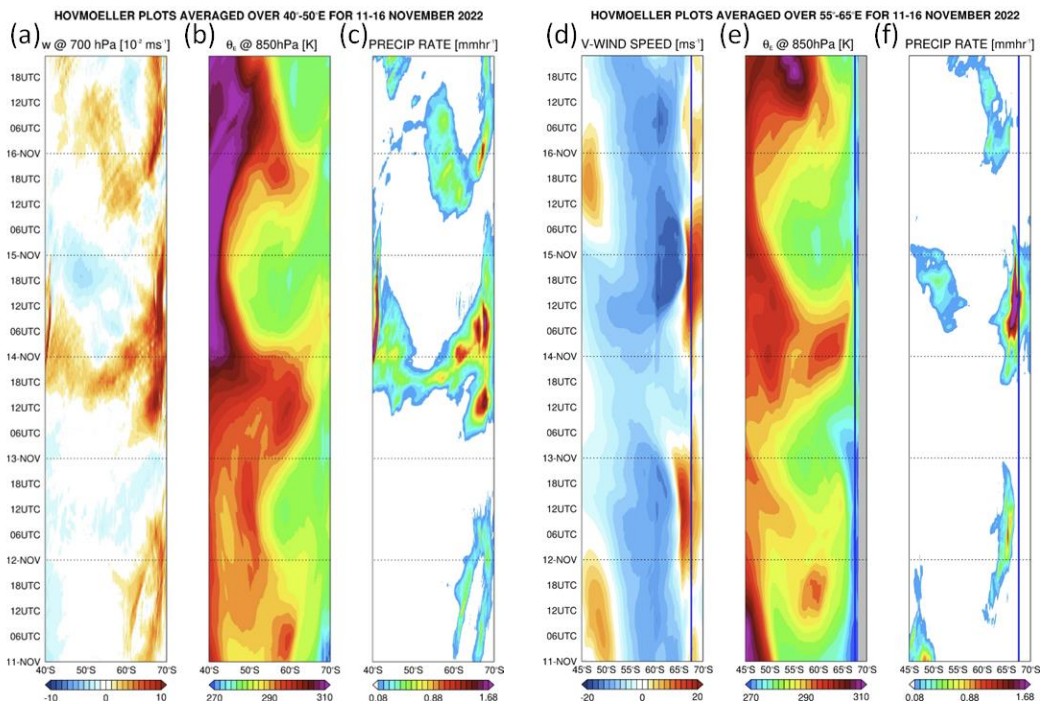

**Figure 9: Hovmoeller Plots:** Hovmoeller of hourly (a) 700 hPa vertical velocity (m s$^{-1}$), (b) 850 hPa equivalent potential temperature (K) and (c) precipitation rate (mm hr$^{-1}$) for 11-16 November 2022 averaged over 40°-50°E, the core of the AR. (d)-(f) are as (a)-(c) but for the (d) 10-m meridional wind speed (m s$^{-1}$), (e) 850 hPa equivalent potential temperature (K) and (f) precipitation rate (mm hr$^{-1}$) averaged over 55°-65°E, where there is a strong interaction between the low-latitude air mass and the katabatic wind flow. The thick blue line in (f) indicates the latitude of the SIMBA site. The grey shading highlights latitudes for which the 850 hPa pressure level is below topography.








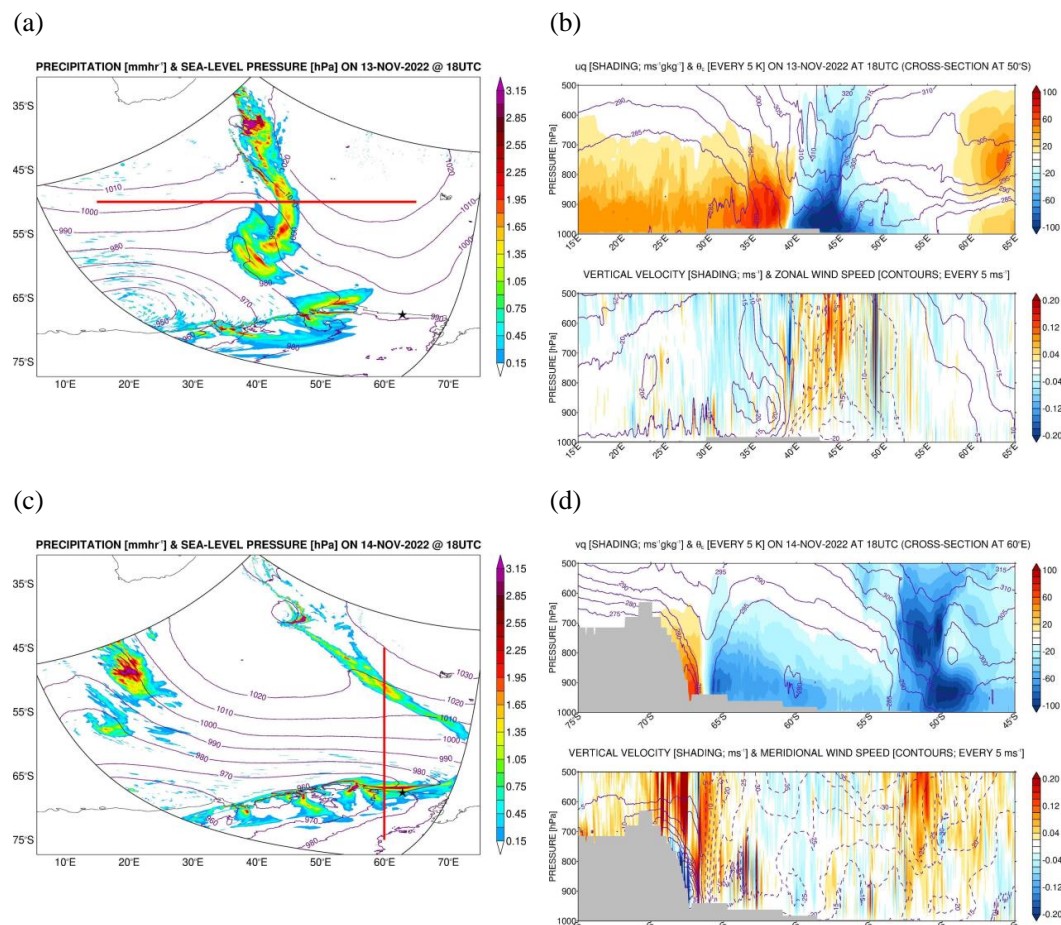

**Figure 10: Precipitation mechanisms in the Southern Ocean:** (a) Precipitation (shading; mm hr$^{-1}$) and sea-level pressure (contours; hPa) at 18 UTC on 13 November 2022, from PWRF's 2.5 km grid. (b) Vertical cross-section at 50°S, red line in (a), of zonal mass transport (shading; m s$^{-1}$ g kg$^{-1}$) and equivalent potential temperature (contours; every 5 K) in the top plot, and vertical velocity (shading; 10$^{-2}$ m s$^{-1}$) and zonal wind speed (contours; every 5 m s$^{-1}$) in the bottom plot, at the same time. Regions below the orography are shaded in grey. (c)-(d) are as (a)-(b) but at 18 UTC on 14 November. The cross-section is at 60°E, with the meridional mass transport and meridional wind speed in the top and bottom plots plotted instead of their zonal counterparts, respectively.


## 5. Discussion and Conclusions

Sea ice is a critically important component of the climate system, modulating atmosphere-
ocean interactions and ultimately the global climate (Raphael et al., 2011; Goosse et al., 2023).
The Antarctic SIE has abruptly dropped from 2016 to2019 (Eayrs et al., 2021; Yang et al., 2021)



with an all time-record low in 2023, driven by a complex and as yet poorly-understood combination of oceanic and atmospheric processes (Wang et al., 2024). Climate model projections indicate major changes in the atmospheric circulation driven by the projected reduction in Antarctic sea ice in a warming climate: the Polar Cell and the katabatic flow off the coast of Antarctica are projected to strengthen, with a marginal weakening of the Ferrel and Hadley cells, and an equatorward shift in the position of the Polar Jet (Tewari et al., 2023). This stresses the need for much improved understanding of the observed variability of sea-ice properties, such as the SIE and SIT that are highly heterogeneous around Antarctica, in order to increase confidence in future climate-change projections.

The SIT at the Khalifa SIMBA site on fast-ice off the Mawson Station largely follows the annual solar (seasonal) cycle, with a gradual increase during winter to mid-to-late October followed by a steady decline in late spring. The maximum values of ~1 m are in the 0.50-1.50 m range estimated from satellite altimeter products for fast-ice in the region around the Mawson Station (Li et al., 2022), and are also comparable to the thickness of pack ice around Antarctica (Kurtz and Markus, 2012; Kacimi and Kwow, 2020). The ST, on the other hand, is highly variable, with values in the range 0.02-0.18 m; these are also consistent with the estimates from the satellite altimetry. In contrast to SIT, the temporal variability of ST is strongly linked to atmospheric forcing, and in particular to episodic warm and moist air intrusions. During July-November 2022, three ARs impacted the site i.e., on 14 July, 13 August and 14 November.  A comparison of reanalysis data with in-situ observations revealed a variation of up to 0.06 m in ST and SIT in response to ARs in both July and August. The warm and moist air masses associated with ARs have a larger impact on sea ice in the colder months, as in the summer the increases in the heat fluxes are partially offset by a decrease in the downward shortwave radiation flux (Liang et al., 2023). The ST and SIT response to the AR occurred within 2 days of its arrival, followed by a recovery to pre-AR levels in the following 2-4 days. However, it is important to stress that a longer observational period (than the current 5-month record) would be needed to establish more robust and statistically significant links between incursions of warm and moist air from low-latitudes and coastal SIT and ST. The air temperature exhibited a marked increase of up to 18 K within 24 h at the site in the case of the 14 July AR, with a less pronounced effect in the summer months (3 K). The in-situ snow, sea-ice and temperature observations highlight the, at times, strong response in particular to ARs impacting the site.

The 14 November AR was particularly intense, with the highest IVT of $161\,\mathrm{kg\,m^{-1}\,s^{-1}}$. From 14 to 15 November, there is a 0.06 m increase in ST and 0.04 m increase in SIT, followed by a return to pre-AR levels on 19 November for SIT and 20 November for ST. The increase in SIT can be explained by the freezing of (some of) the snow on top of the sea-ice, during a time when the surface and air temperatures were below freezing at the site. The period 10-19 November 2022 is characterized by a strong positive SAM phase, with the SAM index being more than 1.5 standard deviations above the 1979-2021 climatological mean, in line with an ongoing La Niña. A pressure



dipole, with a low to the west and a ridge to the east, promotes the advection of warm and moist
low-latitude air across the Mawson Station, with the IVT values in the top 1% of the 1979-2021
climatological distribution and air temperature anomalies in excess of 8 K or more than two
standard deviations above the 1979-2021 mean in parts of East Antarctica between 0º and 70ºE. A
back-trajectory analysis indicates that contributions from evaporation both in the subtropics and
the Southern Ocean contributed to the precipitation event on 14 November 2022. More in-depth
analysis reveals that a secondary low formed just northwest of the site on 14 November, driven by
highly baroclinicity arising from the interaction of the warmer low-latitude air masses with cold
katabatic winds that prevail around the Mawson Station. At the same time, a TPV and a jet streak
at upper-levels contributed to the intensification of the primary low to the west. The changing wind
field also has an impact on the sea-ice dynamics in the region, with maximum pack-ice drift
velocities in excess of 25 km day$^{-1}$ north of the Mawson Station from 14-16 November, an order
of magnitude larger than the 2.5 km day$^{-1}$ during 12-14 November 2022.
A high-resolution simulation with PWRF down to 2.5 km is conducted to gain further insight
into this event. An evaluation against in-situ observations indicated a good performance for both
coastal and inland stations in the target region. A dry bias at coastal sites is attributed to an
excessive offshore wind direction in the model, while at Syowa Station, for which surface radiation
fields are available for evaluation, an underestimation of the upward shortwave radiation flux may
be a reflection of a lower albedo in the model. Regarding the latter, and for 11-16 November 2022,
the surface albedo in PWRF is typically 10% lower than that observed. This suggests the need to
optimize the land surface properties in PWRF, as has been highlighted by other studies such as
Hines et al. (2019), which will be left for future work. Ingesting a more realistic representation of
the SIE and SIT does not translate into higher skill scores for this particular event. This suggests
that improvements to the boundary layer dynamics and/or land/ice processes, highlighted by
studies such as Wille et al. (2016, 2017) and Vignon et al. (2019), and at least for the case study
considered here, are probably more important than having a more accurate sea-ice representation
in the model. In contrast to a September 2017 AR over Greenland (Box et al., 2023) and an April
2023 AR in the Arabian Peninsula (Francis et al., 2024), AR rapids are not seen for this particular
event. The high-resolution model simulations highlight the strong interaction between the air
masses around the low to the west and the high to the east in the Southern Ocean, as well as the
effects of the katabatic wind regime in slowing down and weakening the lower-latitude warm and
moist air incursions as they approach the Antarctic coast. It is the latter interaction that triggers
precipitation rates in excess of 3 mm hr$^{-1}$ around the Mawson Station during 14 November AR,
with the precipitation spatial pattern reflecting the complex topography of the region.
The SIMBA deployment at a fast-ice site off the Mawson Station during July-November 2022
enabled a better understanding of the spatial and temporal variability of SIT and ST in that part of
Antarctica. Such measurements should also be conducted at other sites given the marked regional
differences in sea-ice properties in the Southern Ocean (Parkinson and Cavalieri, 2012). This will





also help to evaluate and improve the ST, SIE and SIT estimates and key products from remote
sensing and numerical models. Besides ocean dynamics and thermodynamics, the findings of the
study stress the role of atmospheric forcing in driving in particular the ST variability. Long-term
measurements are needed to further explore how warm and moist air intrusions modulate the SIT
(not just the SIE) and ST, and how they respond to seasonal and inter-annual variations in the
atmospheric and oceanic state. This is a crucial step to improve the quality and confidence of future
climate change projections and medium- and long-range weather forecasts owing to the global
effects of sea-ice variability on the climate system.

## Acknowledgements

The authors wish to acknowledge the contribution of Khalifa University's high-performance
computing and research computing facilities to the results of this research. The SIMBA
deployment at a fast-ice site off the Mawson Station, *in situ* measurements and the technical
assistance were supported under Australian Antarctic Science [AAS] project #4506 (CI: P. Heil).
The work of R. Massom was supported by the Australian Antarctic Division. For R. Massom, this
work was also supported by the Australian Research Council Special Research Initiative the
Australian Centre for Excellence in Antarctic Science (Project Number SR200100008). PH
acknowledges support from the AAS Program (AAS4496, AAS4506, AAS4625) and grant
funding from the International Space Science Institute (Switzerland; Project 405) and the Swiss
Federal Research Fellowship program. This work contributes to Project 6 of the Australian
Antarctic Program Partnership (ASCI000002) funded under the Australian Government's
Antarctic Science Collaboration Initiative program. We are also grateful for the Byrd Polar and
Climate Research Center at The Ohio State University for developing and maintaining PolarWRF
and making it freely available to the scientific community. We greatly appreciate the support of
the Automatic Weather Station Program and the Antarctic Meteorological Research Center for the
weather station data used in this study (National Science Foundation grants numbers ARC-
0713843, ANT-0944018, and ANT-1141908). The authors also gratefully acknowledge the
National Oceanic and Atmospheric Administration Air Resources Laboratory for the provision of
the Hybrid Single-Particle Lagrangian Integrated Trajectory (HYSPLIT) transport and dispersion
model used in this work.

## Code/Data availability


The sea-ice and snow thickness measurements at the Khalifa SIMBA site on fast-ice off the
Mawson Station for July-November 2022 are available upon request from the corresponding
author (Diana Francis; diana.francis@ku.ac.ae). The remaining observational and the reanalysis
datasets used in this study are freely available online: (i) ERA-5 reanalysis data were downloaded
from the Copernicus Climate Data Store website (Hersbach et al., 2023a,b); (ii) Automatic



Weather Station (AWS) data at the Mawson Station can be requested at the Australian Antarctic Data Center website (AADC, 2022); (iii) AWS and surface radiation data for Syowa Station were obtained from the World Radiation Monitoring Center - Baseline Surface Radiation Network website (AWI, 2024); (iv) AWS data for the Mizuho and Relay stations were extracted from the Antarctic Meteorological Research Center & Automatic Weather Stations Project (Lazzara, 2024); (v) true colour visible daily satellite images from the measurements collected by the Moderate Resolution Imaging Spectroradiometer instrument onboard the Terra satellite were accessed on the National Aeronautics and Space Administration's Worldview website (Boller, 2024); (vi) sea-ice velocity vectors from the low resolution sea-ice drift product are available at the European Organization for the Exploitation of Meteorological Satellites (EUMETSAT) Ocean and Sea Ice Satellite Application Facility (EUMETSAT, 2024); (vii) sea-ice concentration maps derived from the measurements collected by the Advanced Microwave Scanning Radiometer 2 instrument onboard the Japan Aerospace and Exploration Agency Global Change Observation Mission 1st-Water "Shizuku" satellite from January 2013 to present, were obtained from the University of Bremen website (UoB; 2024); (viii) sounding profiles from Syowa Station were accessed at the University of Wyoming website (Oolman, 2024). The Hybrid Single-Particle Lagrangian Integrated Trajectory (HYSPLIT) transport and dispersion model is downloaded from the National Aeronautic and Space Administration Air Resources Laboratory website (NOAA ARL, 2024). The PolarWRF model version 4.3.3 is available at the Byrd Polar and Climate Research Center at The Ohio State University website (PWRF, 2024). The figures presented in this manuscript have been generated with the Interactive Data Language (IDL; Bowman, 2005) and MATLAB (Mathworks, 2024) software.

## Competing interests

One co-author is a member of The Cryosphere editorial board.

## Author Contributions: CRediT

**DF**: Conceptualization of the study, Interpretation and validation of the results, Writing the draft, Funding Acquisition; **RF**: Formal analysis, Data processing and analysis of the results, Writing the draft; **NN**: Data acquisition, processing and analysis, Interpretation of the results, Inputs to the manuscript; **PH**: Interpretation of the results, Inputs to the manuscript; **JDW**: Interpretation of the results, Inputs to the manuscript; **IVG**: Interpretation of the results, Inputs to the manuscript; **RAM**: Interpretation of the results, Inputs to the manuscript. All authors interpreted the results and provided input to the final manuscript.



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
