# Peer review of "Drivers of Observed Winter-Spring Sea-Ice and Snow Thickness at a"

_EGUsphere, 2024_

## Referee Comment (RC1)

**Review of "Impacts of Atmospheric Dynamics on Sea-Ice and Snow Thickness at a Coastal Site in East Antarctica"**

The paper explores complex interactions between atmospheric dynamics and sea-ice/snow thickness variability at a coastal East Antarctic site near Mawson Station. Using in situ SIMBA buoy measurements from July to November 2022, combined with atmospheric reanalysis and PolarWRF modeling, the authors identify key factors driving changes in sea-ice and snow thickness. Main findings highlight the dominant influence of the seasonal solar cycle on sea-ice thickness, while atmospheric processes such as cyclonic forcing, katabatic winds, and atmospheric rivers (ARs) significantly impact snow thickness variability. Overall, the study provides valuable observational insights and model analyses, essential for refining Antarctic climate projections.

I appreciate the comprehensive sea ice observations and the detailed use of PolarWRF. However, several key issues related to model implementation, representation of the chosen site, and methods used for data analysis and interpretation need to be addressed. Consequently, I recommend major revisions before the manuscript can be considered for publication.

**General Comments**:

1. The introduction is more like thesis general introduction instead of the scientific paper including clear motivation and contectual logic. For example, authors selected Mawson Station, but the introduction lacks a robust justification explaining why this particular site is chosen. Is it representative? Is it an area significantly influenced by atmospheric rivers or other dynamic processes compared to other coastal sites? Secondly, it shifts somewhat abruptly between general background, specific processes (e.g., atmospheric rivers and katabatic winds), and observational/modeling studies without clear transitions. Thirdly, one of the major flaws is the absence of explicitly stated research questions or clear objectives in the introduction. The introduction does not adequately highlight the novelty or unique contribution of this particular research. How does this work build upon previous studies, and what new insights does it aim to provide?

2. Data section: Again, the narrative style, while detailed, is more typical of a report than a concise scientific paper, making it harder for me to quickly grasp which datasets were used and their specific purpose. Also, the description of datasets (e.g., SIMBA, ERA-5, AMSR, MODIS, AWS data, sounding profiles) is embedded in paragraphs without clear structuring or consistent formatting, please use table to make your scientific question more clearly.

3. Equations (1)-(5) detailing verification are presented in substantial detail, and they should be better placed in a supplementary materials section. Equestion 7-9, and I am not sure why authors want to list every details in the method part since all different method will compromise the focus of the paper and lose direction, in which some of them should also be put into supplementary. For example, the identification of TTT events can provide valuable context, in this study no significant TTT event was identified during the main AR episode (mid-November 2022). The detailed explanation and equations for the TTT index thus add complexity without substantially advancing core analysis. The same problem is also existing in TPV trackling, which is relatively peripheral to the primary observational focus.

4. PolarWRF description. This part needs significant improvement. Although ERA-5 reanalysis data are mentioned as boundary conditions, details about precisely which variables are prescribed or nudged are scattered and unclear. Additionally, the distinction between default and adjusted sea-ice concentration/thickness ("PWRF" vs. "PWRF_SIE_SIT") and their specific forcing sources (ERA-5 versus satellite data) should be explicitly clarified. The overly detailed description of model physics parameterizations overshadow essential information and could be reduced or moved to supplementary material.

5. Line 490: It is very unclear. The authors state "parameterization schemes" are used to calculate surface sublimation and blowing snow sublimation/divergence, but these are not explicitly defined or cited. The authors should clearly state the specific parameterizations used, are these internal PolarWRF parameterizations or externally applied?

6. SIMBA sea ice thickness and snow depth deduction: While thresholds for distinguishing interfaces (air-snow, snow-ice, ice-water) are provided (lines 229-238), but uncertainties in thickness estimations arising from the threshold are not clearly explored. Although the authors mention initial manual measurements of snow thickness, sea ice thickness, and freeboard at deployment (lines 206-212), there is no clear mention or detailed presentation of subsequent manual validations or calibrations. For example, identifies air-snow and snow-ice interfaces based solely on thermistor temperature gradients after heating, then how is the potential error sources from flooding or snow-to-ice transformations in the event like AR-induced snowfall?

7. AR detection (line 389-392): the authors chose MERRA-2 instead of ERA-5 to identify ARs (lines 389-392), but this choice is not justified clearly.

8. Linking AR and sea ice response: While the authors suggest a clear association between AR occurrences and changes in snow depth, the analysis is primarily qualitative. For instance, they claim a response of about 0.06 m in ST to ARs, but I don't see how robust these associations are statistically. The discussion of AR effects on SIT is even more speculative, especially given the minimal observed changes (0.04 m), which may be within the noise or SIMBA measurement uncertainty. Also, current analysis relies heavily on visual inspection of figures (Fig. 2 and 3), I feel it is difficulty to definitively attribute observed snow and ice changes directly to AR events, given the influence of other processes such as oceanic forcing, seasonal variability, or katabatic winds. More importantly, although some hypotheses are stated regarding snow-to-ice metamorphism and potential snow-ice formation (lines 528-538), can you find the evidences for that?

9. The authors briefly discuss katabatic and foehn winds but don't fully disentangle these from AR-driven changes (lines 522-527). There is insufficient clarity on whether the snow depth changes are primarily due to AR-driven snowfall or secondary processes such as blowing snow removal and sublimation, making conclusions unclear.

10. The authors mention multiple atmospheric phenomena (ARs, katabatic winds, foehn winds), but it remains unclear how confidently observed thickness changes can be attributed exclusively to AR events versus other atmospheric or local processes (katabatic winds, sublimation, blowing snow, ice deformation).

11. In section 4, it is still not adequately demonstrated whether observed precipitation and snow depth changes are uniquely caused by AR-driven moisture transport or influenced by local katabatic processes or other dynamics. Since the authors are using WRF, please consider the appropriate sensitivity simulations using PolarWRF with and without AR-induced moisture to explicitly isolate the contribution of AR moisture to observed snowfall and resulting snow depth and SIT changes. The current case study overly attributes observed sea-ice and snow changes primarily to the AR, without sufficiently considering alternative explanations such as local ocean-ice processes, ice dynamics (e.g., deformation), or blowing snow processes that might have simultaneously influenced SIT and snow depth.

12. PolarWRF: (1) The authors recognize a persistent dry bias and overly strong boundary layer mixing (lines 739-748), the authors do not adequately discuss how such model limitations specifically impact their ability to quantify AR-driven effects on sea ice and snow. (2) Although model performance is evaluated against AWS data, there is no clear, quantitative comparison between PolarWRF-derived snowfall and the actual snow accumulation measured by SIMBA.

13. Discussion: (1) The authors again acknowledge the limited observational period (July–November 2022), noting its inadequacy for statistically robust conclusions (lines 860-862). While this limitation is mentioned, the authors should clearly explain here how future studies or additional observations could specifically address these gaps. (2) since the previous sections has already identified some weakenss, e.g., boundary-layer dynamics, excessive mixing, surface albedo issues, and sensitivity to sea ice representation, it is no clear recommendation or acknowledgment on precisely how to address these issues in future research. (3) The discussion does not clearly revisit the identified uncertainties or assumptions regarding SIMBA buoy-derived SIT and snow depth.

**Specific Comments:**

Check carefully for spacing and punctuation errors throughout the manuscript, especially in lines 522 and 834, where incomplete sentences or extra dots are present.

---

## Author Comment (AC1)

**Reviewer 1:**

The paper explores complex interactions between atmospheric dynamics and sea-ice/snow thickness variability at a coastal East Antarctic site near Mawson Station. Using in situ SIMBA buoy measurements from July to November 2022, combined with atmospheric reanalysis and PolarWRF modeling, the authors identify key factors driving changes in sea-ice and snow thickness. Main findings highlight the dominant influence of the seasonal solar cycle on sea-ice thickness, while atmospheric processes such as cyclonic forcing, katabatic winds, and atmospheric rivers (ARs) significantly impact snow thickness variability. Overall, the study provides valuable observational insights and model analyses, essential for refining Antarctic climate projections.

I appreciate the comprehensive sea ice observations and the detailed use of PolarWRF. However, several key issues related to model implementation, representation of the chosen site, and methods used for data analysis and interpretation need to be addressed. Consequently, I recommend major revisions before the manuscript can be considered for publication.

REPLY: We would like to thank the reviewer for taking the time to go through the manuscript in detail and making multiple constructive comments/suggestions that help to substantially improve its quality. Below we list all the reviewer's inputs and reply to each of them separately, highlighting where changes, if any, are made in the text.

**General Comments:**

1. The introduction is more like thesis general introduction instead of the scientific paper including clear motivation and contextual logic. For example, authors selected Mawson Station, but the introduction lacks a robust justification explaining why this particular site is chosen. Is it representative? Is it an area significantly influenced by atmospheric rivers or other dynamic processes compared to other coastal sites? Secondly, it shifts somewhat abruptly between general background, specific processes (e.g., atmospheric rivers and katabatic winds), and observational/modeling studies without clear transitions. Thirdly, one of the major flaws is the absence of explicitly stated research questions or clear objectives in the introduction. The introduction does not adequately highlight the novelty or unique contribution of this particular research. How does this work build upon previous studies, and what new insights does it aim to provide?

REPLY: We thank the reviewer for his/her comments on the Introduction of our study. We have made substantial changes to the text in the revised version, both to structure it in a more logical way and ensure a smooth flow between the different paragraphs/topics. The first paragraph gives a general background, where the effects of sea-ice on the atmosphere-ocean coupled system (and vice-versa) are noted. In the second paragraph, the focus is on the atmospheric processes that are known to have a major effect on the Antarctic sea ice-snow system. At this stage, Atmospheric Rivers (ARs) and their effects are introduced. We then mention the SIMBA instrument and state why the Mawson Station is selected for its deployment: as now clearly stated in the text (lines 110-112), this area has amongst the highest AR frequency anywhere in Antarctica, with a statistically significant increase in AR frequency and intensity during 1980-2020 (Wille et al., 2025). In the following paragraph we discuss SIE/SIT/ST studies, stressing the importance of measuring them *in-situ*, as these observations can be used to calibrate and evaluate satellite-derived and modelled products. We then have a paragraph outlining the objectives of the study, now made abundantly clear (lines 163-175), and how this work addresses existing gaps in the current understanding of the SIT and ST variability in coastal Antarctica. This is followed by a brief summary of the structure of the article. We believe the overall structure and readability of the Introduction are much improved, and would like to thank the reviewer again for his/her feedback on it.

2. Data section: Again, the narrative style, while detailed, is more typical of a report than a concise scientific paper, making it harder for me to quickly grasp which datasets were used and their specific purpose. Also, the description of datasets (e.g., SIMBA, ERA-5, AMSR, MODIS, AWS data, sounding profiles) is embedded in paragraphs without clear structuring or consistent formatting, please use table to make your scientific question more clearly.

REPLY: We apologize for the lack of clarity regarding the data products used in this work. Following the reviewer's suggestion, we have added a table (Table 1) in which we list all observational and reanalysis datasets used in this study. We have also rephrased the text and structured it in a more logical way: one paragraph for the satellite-derived products (lines 236-248), one paragraph for the ground-based and sounding observations (lines 250-259), and one paragraph for the ERA-5 reanalysis data (lines 261-264). We believe the readability of this section is much improved following the reviewer's feedback.

3. Equations (1)-(5) detailing verification are presented in substantial detail, and they should be better placed in a supplementary materials section. Equestion 7-9, and I am not sure why authors want to list every details in the method part since all different method will compromise the focus of the paper and lose direction, in which some of them should also be put into supplementary. For example, the identification of TTT events can provide valuable context, in this study no significant TTT event was identified during the main AR episode (mid-November 2022). The detailed explanation and equations for the TTT index thus add complexity without substantially advancing core analysis. The same problem is also existing in TPV trackling, which is relatively peripheral to the primary observational focus.

REPLY: We thank the reviewer for his/her comments on the methodology section of our article. We do agree it is rather long, and that some parts can be relegated as supplementary material, including the verification diagnostics and metrics used to diagnose the Southern Annular Mode (SAM), atmospheric blocking, tropical temperate troughs (TTTs), and tropopause polar vortices (TPVs). The TTTs are removed from the manuscript as they are not relevant for the new case study selected for modeling, 11-16 July 2022, in light of the change to the AR detection methodology in response to the reviewer's general comment #7. We have followed the reviewer's suggestion in the revised version, with only a brief reference (no more than 1-2 sentences) in the main text (lines 318-324 and 335-338) with the technical details given in Supplement Sections S1-S3.

4. PolarWRF description. This part needs significant improvement. Although ERA-5 reanalysis data are mentioned as boundary conditions, details about precisely which variables are prescribed or nudged are scattered and unclear. Additionally, the distinction between default and adjusted sea-ice concentration/thickness ("PWRF" vs. "PWRF_SIE_SIT") and their specific forcing sources (ERA-5 versus satellite data) should be explicitly clarified. The overly detailed description of model physics parameterizations overshadows essential information and could be reduced or moved to supplementary material.

REPLY: We agree with the reviewer's feedback and have revised section 2.3 accordingly. In particular, the physics schemes are now listed in Table 2 and have been removed from the text. We now have one paragraph describing the model configuration and interior nudging details (lines 287-298), explicitly stating which variables are nudged, at which levels, and the nudging time-scale. A second paragraph focuses on the sensitivity experiments (lines 303-312), explaining why "PWRF_SIE_SIT" was conducted and how the SIE and SIT in this run differ from those in the control ("PWRF") simulation.

5. Line 490: It is very unclear. The authors state "parameterization schemes" are used to calculate surface sublimation and blowing snow sublimation/divergence, but these are not explicitly defined or cited. The authors should clearly state the specific parameterizations used, are these internal PolarWRF parameterizations or externally applied?

REPLY: We apologize for the lack of clarity regarding the way the surface sublimation/evaporation rate, blowing snow sublimation rate, and blowing snow divergence are parameterized. We have referenced Francis et al. (2023) in which all the details are given. However, and in light of the reviewer's comment, we have added a subsection to the supplementary with the relevant equations (Supplement Section S4). The text in the main article has been updated to reflect this (line 360-363). In addition, and as we now make clear in the text (lines 363-365), these equations are used to estimate three terms of the SMB using hourly data from ERA-5 and PWRF with the results displayed in Figs. 3 and S5, respectively. They are not coded in the model.

6. SIMBA sea ice thickness and snow depth deduction: While thresholds for distinguishing interfaces (air-snow, snow-ice, ice-water) are provided (lines 229-238), but uncertainties in thickness estimations arising from the threshold are not clearly explored. Although the authors mention initial manual measurements of snow thickness, sea ice thickness, and freeboard at deployment (lines 206-212), there is no clear mention or detailed presentation of subsequent manual validations or calibrations. For example, identifies air-snow and snow-ice interfaces based solely on thermistor temperature gradients after heating, then how is the potential error sources from flooding or snow-to-ice transformations in the event like AR-induced snowfall?
REPLY: We agree with the reviewer it is essential to provide the uncertainties in the estimations of SIT and ST so as to better quantify whether their changes are statistically significant. Following the reference article for the methodology considered here, Liao et al. (2018), the uncertainty in ST is estimated to be 2-7% and that in SIT is estimated to be 1-2%. Hence, we take 7% for ST and 2% for SIT and highlight the range in Figs. 3e-f, 3k-l, and S5e-f. This is also mentioned in the text (lines 374-375). The Khalifa SIMBA instrument is manually validated at the beginning of the measurement campaign, with no calibration or validation during the 07 July – 07 December period. We also state this in the text (lines 193-194). We would also like to highlight that the field campaign took place from mid-winter to late autumn, when flooding, snow-ice formation, and snow melting are less likely compared to early winter and summer (as seen in Fig. 2a, the surface and air temperature at the site remained below freezing during July-November 2022). Given this, the uncertainty of 7% for ST and 2% for SIT is deemed representative of the error sources during the measurement period.

7. AR detection (line 389-392): the authors chose MERRA-2 instead of ERA-5 to identify ARs (lines 389-392), but this choice is not justified clearly.
REPLY: We would like to thank the reviewer for raising the issue of the reanalysis dataset used for the identification of ARs. We used MERRA-2 as our initial goal was to investigate aerosol atmospheric rivers and their effects on sea-ice and snow around Antarctica. However, and at least for the July-November 2022 period, we found the amount of aerosols that reached the Mawson Station was very small, and hence our study was directed towards ARs and their impacts on the cryosphere. Given this, we agree it does not make sense to use MERRA-2 just for AR identification and ERA-5 for all other analyses. In the revised version of the manuscript we have used ERA-5 integrated vapour transport (IVT) to identify ARs (lines 330-334), meaning only one reanalysis dataset is considered in this work. With the updated methodology, we only have one intense AR during the study period, on 14 July 2022, which was selected for the modelling work. Section 4 was therefore updated accordingly, now targeting this event.

8. Linking AR and sea ice response: While the authors suggest a clear association between AR occurrences and changes in snow depth, the analysis is primarily qualitative. For instance, they claim a response of about 0.06 m in ST to ARs, but I don't see how robust these associations are statistically. The discussion of AR effects on SIT is even more speculative, especially given the minimal observed changes (0.04 m), which may be within the noise or SIMBA measurement uncertainty. Also, current analysis relies heavily on visual inspection of figures (Fig. 2 and 3), I feel it is difficulty to definitively attribute observed snow and ice changes directly to AR events, given the influence of other processes such as oceanic forcing,

seasonal variability, or katabatic winds. More importantly, although some hypotheses are stated regarding snow-to-ice metamorphism and potential snow-ice formation (lines 528-538), can you find the evidences for that?

REPLY: We thank the reviewer for pointing out the lack of quantitative statements regarding the role of different atmospheric processes on ST and SIT variability. We have improved Fig. 3 by adding additional fields to better identify the different phenomena and quantify their impact on ST and SIT. As we now state in the text (lines 400-416 and 418-440), precipitation (snowfall), Foehn effects, blowing snow sublimation, and episodic warm and moist air intrusions modulate the ST by up to 0.08 m in a day. On the other hand, the variability in SIT does not appear to be linked to atmospheric forcing: e.g. the 0.02 m variations during the passage of the 14 July 2022 AR are within the measurement uncertainty. We have removed the references to snow-to-ice metamorphism and potential snow-ice formation mechanisms from the text as we cannot back them up with the available data. The discussion of Figs. 2 and 3 has been revised following the reviewer's input. Worth mentioning that our study highlights the fact that AR events are not synonym of net snow accumulation and that this depends on other factors such as winds and temperature. Without in-situ measurements this couldn't be discovered.

9. The authors briefly discuss katabatic and foehn winds but don't fully disentangle these from AR-driven changes (lines 522-527). There is insufficient clarity on whether the snow depth changes are primarily due to AR-driven snowfall or secondary processes such as blowing snow removal and sublimation, making conclusions unclear.

REPLY: We apologize for the lack of clarity in the article regarding the potential role of Foehn effects on snow depth. In light of the reviewer's comment, we have revised the full paragraph (lines 385-440) and added additional fields to Fig. 3 (air temperature, relative humidity, horizontal wind direction and speed) to back-up our statements. Both precipitation events, blowing snow, Foehn effects, and low-latitude air intrusions play a role in the variability of ST during the study period. In very windy conditions, such as in the 14 July AR, ST may not increase as the snow is blown away as quickly as it falls, as has been reported at the Mawson (and neighbouring) sites by some studies. When the wind speed is low, precipitation typically accumulates, and the ST increases by up to 0.06 m. Foehn effects, on the other hand, can lead to a decrease in ST by as much as 0.08 m. Blowing snow sublimation also modulates ST, and can lead to a decrease by up to 0.08 m, with blowing snow divergence playing a much reduced role. The surface and air temperature remained below freezing at the Khalifa SIMBA site during the study period, and therefore surface melting is not expected to play a role in the ST variability, as evidenced by the SMB analysis. As the reviewer can see, we now present a much-improved discussion of the atmospheric processes driving the ST variability in the text, identifying them through the analysis of relevant meteorological variables, and quantifying their impact, including for the 14 July 2022 AR passage (lines 400-416 and 418-436).

10. The authors mention multiple atmospheric phenomena (ARs, katabatic winds, foehn winds), but it remains unclear how confidently observed thickness changes can be attributed exclusively to AR events versus other atmospheric or local processes (katabatic winds, sublimation, blowing snow, ice deformation).

REPLY: As mentioned in the reply to the reviewer's previous comment, we have rewritten the discussion on the atmospheric processes that modulate ST, identifying them with relevant meteorological data and quantifying their role in the ST variability (lines 385-440). Snowfall events, if not accompanied by very strong winds as in ARs, generally lead to an increase in ST by up to 0.06 m, whereas Foehn effects and blowing snow can lead to variations of up to $\pm$ 0.08 m and + 0.08 m, respectively. Sublimation is unlikely to have played a role in the ST during July-November 2022 as the surface and air temperature remained below freezing. As the reviewer would agree, the discussion in section 3 is much improved in the revised version of the manuscript. We would like to thank him/her for raising the issue of a lack of clarity in our presentation of the atmospheric phenomena that drives ST variability during the study period.

11. In section 4, it is still not adequately demonstrated whether observed precipitation and snow depth changes are uniquely caused by AR-driven moisture transport or influenced by local katabatic processes or other dynamics. Since the authors are using WRF, please consider the appropriate sensitivity simulations using PolarWRF with and without AR-induced moisture to explicitly isolate the contribution of AR moisture to observed snowfall and resulting snow depth and SIT changes. The current case study overly attributes observed sea-ice and snow changes primarily to the AR, without sufficiently considering alternative explanations such as local ocean-ice processes, ice dynamics (e.g., deformation), or blowing snow processes that might have simultaneously influenced SIT and snow depth.

REPLY: We thank the reviewer for raising the important issue of exploring the role of different processes on ST and SIT changes rather than just attributing them to the effects of the AR. As stated in the reply to the reviewer's general comments #8-10, we have rewritten section 3 of the article and now highlight the role of blowing snow, Foehn effects, and non-AR precipitation on the ST changes (lines 385-440). For the case study, 11-16 July, the close agreement between the ERA-5 and PWRF SMB budget confirms the role of Foehn effects and the distinct impact of AR and non-AR precipitation on ST (lines 651-662). As we now make clear in the text (lines 436-438), the 0.02 m variations in SIT during the AR passage are within the uncertainty range, and hence SIT did not change during the event. We would like to note that PWRF is run without an ocean and sea-ice model, and hence the role of ice dynamics and local ocean-ice processes cannot be investigated with the model output. Coupling PWRF with ocean and sea-ice models is outside the scope of this study. In light of the additional analysis performed with the PWRF output, the fact that the model does not simulate ocean and sea-ice dynamics, and that the new case study takes place during the Polar night with the AR being the major moisture source (the air mass over the Antarctic continent is bone dry with water vapour mixing ratios generally below $0.1 \, g \, kg^{-1}$, as evidenced by the *in-situ* observations at the Relay Station, Fig. S4h), we believe a sensitivity run in which the AR-induced moisture is isolated is not necessary and is unlikely to bring added value to the study. We would also like to stress the high computational cost of the PWRF simulations with the considered model set up: with 1,040 cores it takes roughly 21 days to complete the 7-day simulation. We hope the reviewer understands our decision regarding conducting additional sensitivity simulations.

12. PolarWRF: (1) The authors recognize a persistent dry bias and overly strong boundary layer mixing (lines 739-748), the authors do not adequately discuss how such model limitations specifically impact their ability to quantify AR-driven effects on sea ice and snow. (2) Although model performance is evaluated against AWS data, there is no clear, quantitative comparison between PolarWRF-derived snowfall and the actual snow accumulation measured by SIMBA.

REPLY: Regarding (1), and as we now state in the text (lines 648-651), PWRF captures the effects of the AR as seen in observations, most notably the increase in air temperature, water vapour mixing ratio, and wind speed in particular at the Mawson and Davis stations where the AR impacts are more evident. The dry bias, which arises from a more offshore wind direction, and the likely stronger boundary layer mixing do not preclude PWRF from simulating the 14 July 2022 AR. In fact, the close agreement between the model's and ERA-5's SMB budget (lines 651-662) confirms its ability to capture the AR effects on the ST. Regarding (2), we appreciate the reviewer for pointing this out, we missed it completely! In the revised version we added one panel (Fig. 8f) where the model-predicted and observed ST at the Khalifa SIMBA site are plotted, and discuss the results in the text (lines 664-679).

13. Discussion: (1) The authors again acknowledge the limited observational period (July–November 2022), noting its inadequacy for statistically robust conclusions (lines 860-862). While this limitation is mentioned, the authors should clearly explain here how future studies or additional observations could specifically address these gaps. (2) since the previous sections has already identified some weakenss, e.g., boundary-layer dynamics, excessive mixing, surface albedo issues, and sensitivity to sea ice representation, it is no clear recommendation or acknowledgment on precisely how to address these issues in future

research. (3) The discussion does not clearly revisit the identified uncertainties or assumptions regarding SIMBA buoy-derived SIT and snow depth.

REPLY: We thank the reviewer for his/her suggestions to improve the discussion section of the article. Regarding (1), we now state in the text how additional observations can help to further our understanding of the role of atmospheric phenomena such as Foehn effects, blowing snow, and warm and moist air intrusions on ST/SIT (lines 865-870), with numerical simulations with coupled ocean-atmosphere-sea-ice models complementing the analysis (lines 803-806). Regarding (2), we recommend future PWRF studies to explore other physics schemes and/or optimize the tunable parameters defined inside the selected schemes in an attempt to improve the model performance, on top of ingesting more realistic surface properties (lines 842-851). Regarding (3), and following the reviewer's general comment #6, we now show the uncertainty that arises from the methodology used to estimate SIT and ST (Figs. 3e-f, 3k-l, and S5e-f). In the discussion section, in lines 806-809, we state the need for the development of refined methods to estimate these two variables, in particular as the variations of SIT during atmospheric phenomena such as ARs are within the uncertainty range, preventing a signal from being extracted from the data.

Specific Comments:
Check carefully for spacing and punctuation errors throughout the manuscript, especially in lines 522 and 834, where incomplete sentences or extra dots are present.
REPLY: We thank the reviewer for spotting these typos. We have corrected them (lines 418 and 775) and made sure there are no such further occurrences in the revised version of the manuscript.

REFERENCES:

Francis, F., Fonseca, R., Mattingly, K. S., Lhermitte, S., Walker, C. (2023) Foehn winds at Pine Island Glacier and their role in ice changes. The Cryosphere, 17, 3041-3062. https://doi.org/10.5194/tc-17-3041-2023

Liao, Z., Cheng, B., Zhao, J., Vihma, T., Jackson, K., Yang, Q., Yang, Y., Zhang, L., Li, Z., Qiu, Y., Cheng, X. (2018). Snow depth and ice thickness derived from SIMBA ice mass balance buoy data using an automated algorithm. International Journal of Digital Earth, 12(8), 962–979. https://doi.org/10.1080/17538947.2018.1545877

Wille, J. D., Favier, V., Gorodetskaya, I. V., Agosta, C., Baiman, R., Barrett, J. E., Barthelemy, L., Boza, B., Bozkurt, D., Casado, M., Chyhareva, A., Clem, K. R., Codron, F., Datta, R. T., Duran-Alarcon, C., Francis, D., Hoffman, A. O., Kolbe, M., Krakosvska, S., Linscott, G., Maclennan, M. L., Mattingly, K. S., Mu, Y., Pohl, B., Santos, C. L.-D., Shields, C. A., Toker, E., Winters, A. C., Yin, Z., Zou, X., Zhang, C., Zhang, Z. (2025) Atmospheric rivers in Antarctica. Nature Reviews Earth & Environment, 6, 178-192. https://doi.org/10.1038/s43017-024-00638-7

---

## Author Comment (AC2)

**Reviewer #2:**

Antarctic sea ice and snow play an important role in regulating the global climate system. However, scare observations limit our understanding of atmosphere-sea ice-ocean interaction processes in Antarctic. The aim of this study is to improve our understanding of the temporal evolution of sea ice and snow around East Antarctica based on in-situ observations, reanalysis, and simulations. This study starts from presenting the evolution of sea ice thickness and snow depth by using the SIMBA measurements collected at Khalifa site. Then this study provides a wealth of analysis on atmospheric rivers. Overall, I recommend the publication of this study but suggest that the major revisions are needed before publication.

**REPLY:** We would like to thank the reviewer for going through the manuscript in detail and sharing several constructive comments/suggestions that helped to substantially improve the quality of the work. Below we list them and reply to each separately, highlighting where in the text changes, if any, are made.

My biggest concern is that the transition from observation analysis (Sec. 3) to AR analysis (Sec 4) is vague or invalid. This makes the paper appear as two separate parts, lacking overall coherence. In the part of observation analysis, the result shows that ST increases during ARs due to precipitation, and SIT increases by 0.04 m during the 14 November AR due to snow-ice interactions. But in my opinion, the authors do not fully explain the direct impacts of ARs on changes in ST and SIT. I think the following questions should be answered at least, otherwise, it would be far-fetched to directly connect the extensive analysis of AR in the remaining part of the article.

**REPLY:** We agree with the reviewer that in the original submission the analysis of the observational data collected during July-November 2022 was incomplete and lacked a clear link to the meteorological fields. In addition, and following the comments by reviewer #1, we have used ERA-5 to extract the ARs instead of MERRA-2 and have computed the uncertainty in the ST and SIT estimates. Following this, we only have one intense AR during July-November 2022, on 14 July, and the variations of SIT during the passage of the AR are within the uncertainty range, meaning only ST changed during the event. We found that ST did not increase during the heavy snowfall associated with the AR owing to the strong winds (speeds $>30\,\mathrm{m\,s}^{-1}$) that likely prevented snow's accumulation, as reported in the literature at the Mawson (and nearby) Station during strong wind episodes. In fact, the Foehn effects that followed the AR led to a decrease in ST. We have rewritten the discussion of Figs. 3 (lines 385-440) and 4 (lines 442-463) and updated them, in the case of Fig. 3 by also adding relevant meteorological fields and the uncertainty of the ST and SIT estimates. Section 4 now features the period 11-16 July 2022, and we provide a direct link with the analysis performed in section 3: e.g., we now compare the PWRF-predicted snow thickness with that observed *in-situ* (Fig. 8f) and also conduct the SMB analysis for the model forecasts (Fig. S5). This ensures that Sections 3 and 4 are not disconnected, with the model outputs used to better understand the effects and structure of the AR.

- What are meteorological conditions near the sea ice surface during ARs? How do meteorological conditions affect the SIT and ST variations?

**REPLY:** In light of the reviewer's comment, we have added to Fig. 3 the hourly air temperature, relative humidity, and horizontal wind direction and speed to complement the SMB analysis and to compare with the *in-situ* ST and SIT measurements. We now explicitly discuss the role of the meteorological conditions on ST during the AR passage (lines 418-436; the SIT variations are within the uncertainty range) and Foehn

effects (lines 421-432). The effects of different meteorological phenomena on ST are now quantified (lines 400-416).

- It seems that precipitation plays an important role in affecting ST variations. What are the special features of the increase in ST during ARs compared to the ST increase during other snowfall events?
**REPLY:** As we now highlight in the text (lines 427-432), during the AR we do not see an increase in ST as the strong winds, with speeds in excess of $30\,\text{m}\,\text{s}^{-1}$, likely prevent its local accumulation, as reported to be the case in the literature in windy conditions. During non-AR snowfall events, when the wind speed is low (e.g. on 16 July when it dropped below $2\,\text{m}\,\text{s}^{-1}$) we do see an increase in ST (on this day by 0.02 m; lines 432-436). In this regard, AR and non-AR snowfall events have a different impact on the SMB. Our study highlights the fact that AR events are not synonym of net snow accumulation and that this depends on other factors such as winds and temperature. Without in-situ measurements this couldn't be discovered.

- Why does the SIT only increase during the 14 November AR period, while it does not increase during other ARs?
**REPLY:** When accounting for the uncertainty in the SIT estimates that arise from the methodology used to compute it, we do not see an increase during the passage of the 14 July AR: as stated in the text (lines 436-440), its 0.02 m variation is within the uncertainty range. As we only have one AR during the study period following a change in the methodology (ARs are now extracted using ERA-5's IVT instead of MERRA-2's vIVT; lines 331-334), we have to be careful not to generalize to the passage of ARs. As we note in the text (lines 438-440 and 865-868), a longer measurement period that comprises multiple AR passages would be needed for a robust conclusion of the effects of ARs on ST and SIT to be reached.

On the other hand, I suggest the authors to add the analysis of observed near-surface meteorological elements (e.g., wind speed, wind direction, air temperature and humidity. ) near the observation site and its impacts on ST and SIT variations.
**REPLY:** We would like to thank the reviewer very much for his/her suggestion, which we implemented in Figs. 3 and S5, that allowed for a more insightful and detailed discussion of the effects of different atmospheric processes on the ST and SIT measurements (lines 400-416 and 418-436).

**Specific comments:**

1. Lines 508~509:How do you infer that SIT changes are mainly caused by oceanic forcing? From Figure 2, it can be seen that changes in SIT are mainly controlled by the growth and melting at the bottom, but it cannot be directly attributed to oceanic forcing, as the growth and melting of ice at the bottom is the result of competition between oceanic heat flux and conductive heat flux, and the conductive heat flux also depends on how much energy the ice absorbs from the atmosphere.
**REPLY:** We agree with the reviewer and would like to thank him/her for pointing out the role of the atmospheric forcing on the SIT variations, which we neglected to mention in the text, referring only to the oceanic heat flux. We have rephrased the sentence accordingly (lines 387-390).

2. Lines 514~520:How does equation 10 consider the process of snow-ice transition? This may affect the explanation of changes in ST with SMB.
**REPLY:** The SMB, defined in new equation (3), considers the different sources and sinks of snow, and does not explicitly represent the snow-ice transition process, even though snow-ice processes (e.g.,

conversion of ice to snow) are indirectly accounted for. We have rephrased the sentence the reviewer is referring to (lines 394-400) and have also expanded the discussion of Fig. 3 (lines 385-440), now with the addition of four meteorological fields that allow for a more comprehensive analysis (and quantification) of the effects of atmospheric processes on the ST variability.

3. Line 522:Add a space between "sea-ice" and "SMB", and delete "." before Foehn.
**REPLY:** In the revised version of the manuscript the referred sentence was removed. In any case, we corrected similar typos elsewhere in the text (e.g. in line 775).

4. Figure 3: The line for SMB is always covered by P line, and the line for M is also invisible. It is easy to cause misunderstandings. I suggest redesigning the display of results, perhaps using dual y-axis can solve this problem.
**REPLY:** We thank the reviewer for his/her suggestion. As we now state in the caption of Fig. 3 and in the text (lines 413-414), the snowmelt ($M$) term is zero for the full 08 July - 30 November measurement period. We have experimented with different options and decided to multiply the blowing snow divergence ($D$) term by two in Fig. 3b for easiness of visualization, stating it in the figure legend and caption, instead of using a dual y-axis. We believe the readability of Fig. 3 has been improved following the reviewer's input.

5. Line 540~542: How should I understand the ST is decreasing during blocking high events, but the occurrence of blocking coincides with the the passage of ARs and ARs always lead to an increase of ST as the observations present?
**REPLY:** We apologize for the lack of clarity in the text and have rephrased the sentence accordingly (lines 442-446). As noted in the reply to the reviewer's major concern and in the text (lines 426-436), at the Mawson Station we do not see an increase in ST in association with the passage of the 14 July 2022 AR, the only AR that impacted the site during the study period. In fact, ST decreases because of Foehn effects. Also, having blocking does not necessarily mean there will be ARs, and not all warm and moist air intrusions meet the strict intensity and geometric requirements of an AR. In addition, and following the reviewer's specific comment #6, we use the $40 \, \mathrm{m \, s^{-1}}$ threshold of the Pook Blocking Index to diagnose blocking events (stipple in Figs. 4a and 4d), meaning no blocking events around the site during July-November 2022 (lines 452-456).

6. Figure 4: How to identify the blocking from Figure 4a and 4d?
**REPLY:** Following the definition of the Pook Blocking Index, equation (S6), positive values indicate weaker mid-latitude (50º-60ºS) westerlies and/or anomalous westerlies at lower- (35º-40ºS) and higher- (65º-70ºS) latitudes, and hence a blocked extratropical westerly flow. We use as threshold $40 \, \mathrm{m \, s^{-1}}$ to identify blocking events (stipple in Figs. 4a and 4d). We have updated the discussion in the text accordingly (lines 442-458).

7. Lines 852~853: The evidence is weak to make this conclusion.
**REPLY:** We have rephrased the referred sentence following the revised discussion of Fig. 3. In particular, we now state that the variability in ST is linked to precipitation (snowfall), Foehn effects, blowing snow, and episodic warm and moist air intrusions, which can lead to variations of up to $\pm 0.08 \, \mathrm{m}$ in a day (lines 792-795), and not just to the warm and moist low-latitude air occurrences.

8. Lines 854~856: Only the increase of 0.06 m in ST during the 14 November AR period is given in the result part.

**REPLY:** Following an update to the methodology used to diagnose ARs, we only have one AR during the study period, on 14 July 2022. During this event, the 0.02 m change in SIT is within the uncertainty range, while the up to 0.04 m variation in ST is likely due to Foehn effects and snowfall. We have rephrased the text accordingly (lines 797-798).

9. Lines 865~866: This is contradictory to your statement given in Lines 568~569.

**REPLY:** We agree with the reviewer that this sentence is incorrect and have removed it from the revised version of the manuscript.

---

## Author Response (AR2)

**Reply to Reviewers:**

We would like to thank the two reviewers for their valuable comments and suggestions in this second round of revision, which helped to further improve the quality of our study. Kindly find below in blue our response point-by-point to their inputs.

**Reviewer #1:**

**The authors have attempted to address all of my comments and the manuscript is improved from the initial submission. However, I think some points of clarification are still needed.**
REPLY: We are glad to know the reviewer is happy with the changes made to the manuscript in the previous round of revision considering his/her feedback. Below we list the further comments/suggestions made and reply to each separately, highlighting where in the text modifications are implemented.

**1. In my opinion, the SIMBA observations show a measly impact of AR on SIT and ST and the simulation results also fail to reproduce the ST variation. But this study does provide a thorough analysis in AR, including large-scale atmospheric patterns and PolarWRF simulation. Hence, please consider if the title of this article is appropriate and weakening the analysis of SIT and ST variations.**
REPLY: In light of the reviewer's comment, we have rephrased the title to "*Drivers of Observed Winter-Spring Sea-Ice and Snow Thickness at a Coastal Site in East Antarctica*". As the reviewer points out, any explicit reference to atmospheric rivers in the title is not justified given the findings of the study, and the mention of atmospheric dynamics is also not appropriate as we partially attribute the changes in SIT to ocean forcing (lines 381-384). The new title just mentions the study delves into the mechanisms behind the variability of the *in-situ* measured SIT and ST at a coastal site in East Antarctica, which is the central aim of the work. The Abstract has also been rephrased to highlight this (lines 24-26).

**2. The relationship between AR, Foehn winds, and katabatic winds should be clear stated. Because the authors sometimes use Foehn winds to explain the variation of ST and sometimes use katabatic winds to explain it.**
REPLY: We thank the reviewer for raising this important issue. We have considered the methodology proposed by Francis et al. (2023) to identify Foehn events during our study period of July - November 2022 (lines 359-365). In particular, and at a given grid-point, a timestamp is denoted as a Foehn timestamp if the 2-m temperature exceeds its $60^{th}$ percentile, the 2-m relative humidity drops below its $30^{th}$ percentile, and the 10-m wind speed exceeds its $60^{th}$ percentile. For the 2-m temperature, monthly hourly thresholds are used to account for the annual cycle, while for the other variables the percentiles are extracted for the full period. Foehn timestamps are shaded in purple in Figs. 3d and 3j. As we note in the text (lines 402-403), Foehn effects are distinguished from katabatic winds as they have to meet the criteria above for the three variables. We have updated the discussion of Fig. 3 accordingly (lines 396-399 and 419-422).

**3. The authors attribute the model's inability to predict the variation of ST to less favourable conditions for Foehn events in the model. But I think it is also important for what processes of snow redistribution are included in the model.**
REPLY: The reviewer makes a very good point. Indeed, the land surface model (LSM) used in the PWRF simulations, Noah LSM, features a single snow layer and a simplified representation of the snow accumulation, sublimation, and melting processes (Lim et al., 2022). In contrast, the more sophisticated

Noah LSM with multiparameterization options (Noah-MP), also available in PWRF, includes up to three snow layers, represents the percolation, retention, and refreezing of meltwater within the snowpack, and accounts for snow metamorphism and compaction (Niu et al., 2011). PWRF simulations have shown the Noah-MP gives more skillful predictions over Antarctica compared to the Noah LSM for fields such as 2-m temperature and 10-m wind speed (Xue et al., 2022). We have stated this in the text (lines 672-682) and suggest that future work should include considering a more detailed representation of snow processes in the model (lines 855-857). The need for a higher spatial resolution to better simulate the atmospheric dynamic and thermodynamic processes including the Foehn event and cloud microphysical processes as highlighted by Gilbert et al. (2025) is also noted (lines 666-669 and 857-859).

**REFERENCES:**

Francis, F., Fonseca, R., Mattingly, K. S., Lhermitte, S., Walker, C. (2023) Foehn winds at Pine Island Glacier and their role in ice changes. The Cryosphere, 17, 3041-3062. https://doi.org/10.5194/tc-17-3041-2023

Gilbert, E., Pishniak, D., Torres, J. A., Orr, A., Maclennan, M., Wever, N., Verro, K. (2025) Extreme precipitation associated with atmospheric rivers over West Antarctic ice shelves: insights from the kilometre-scale regional climate modeling. The Cryosphere, 19, 597-618. https://doi.org/10.5194/tc-19-597-2025

Lim, S., Gim, H.-J., Lee, E., Lee, S., Lee, W. Y., Lee, Y. H., Cassardo, C., Park, S. K. (2022) Optimization of snow-related parameters in the Noah land surface model (v3.4.1) using a micro-genetic algorithm (v1.7a). Geoscientific Model Development, 15, 8541-8559. https://doi.org/10.5194/gmd-15-8541-2022

Niu, G.-Y., Yang, Z.-L., Mitchell, K. E., Chen, F., Ek, M. B., Barlage, M., Kumar, A., Manning, K., Niyogi, D., Rosero, E., Tewari, M., Xia, Y. (2011) The community Noah land surface model with multiparameterization options (Noah-MP): 1. Model description and evaluation with local-scale measurements. Journal of Geophysical Research, 116, D12109. https://doi.org/10.1029/2010JD015139

Xue, J., Xiao, Z., Bromwich, D. H., Bai, L. (2022) Polar WRF V4.1.1 simulation and evaluation for the Antarctic and Southern Ocean. Frontiers of Earth Science, 16, 1005-1024. https://doi.org/10.1007/s11707-022-0971-8

**Reviewer #2:**

The author has carefully revised the manuscript, and the improved presentation is clearer. However, the following issues still require further refinement before publication:

REPLY: We are happy to know the reviewer is satisfied with the revised version of the manuscript. Below we list the two lingering issues raised in this second round of revision and reply to each separately, highlighting where in the text changes are made.

(1) Firstly, the introductory statements are overly redundant and should be streamlined. Secondly, the logical flow between paragraphs is unclear. For example, while the third paragraph effectively elaborates on the importance of extreme weather for studying the sea-ice-snow-air coupling system, the fourth paragraph abruptly shifts to discussing buoy observation methods without a smooth transition, making the narrative seem disjointed. The author is advised to either add transitional sentences or restructure the fourth paragraph by first clarifying the research objectives before delving into methodological details.

REPLY: In light of the reviewer's comment, we have added a transitional sentence at the start of the fourth paragraph of the Introduction (lines 103-105) to ensure a logical flow between the third and fourth paragraphs. In particular, we highlight that having *in-situ* measurements of SIT and ST is crucial to understand the effects of the atmospheric forcing on the sea-ice-snow-air coupling system and then proceed to state the goals of the work. We have also streamlined the Introduction by removing sentences that mostly repeat what has been mentioned before and merging those that complement each other, with an overall 10% reduction in the number of words in this section.

(2) The main figure contains several subfigures, but the descriptions of these subfigures in the text exhibit logical jumps, e. g. Figure 3, which may hinder reader comprehension. The author is recommended to describe the subfigures sequentially.

REPLY: We agree with the reviewer and have changed the order of the panels in Figs. 3 and S5 to ensure that each subfigure is discussed sequentially. In both, the top panels show the observed SIT and ST measurements, followed by the different SMB terms, and finally by the meteorological fields (namely air temperature, relative humidity, wind speed and direction). The text has also been updated accordingly (lines 367-436 and 662-675). The order of the panels of Fig. 1 has also been updated to ensure a sequential discussion in the text. We believe the presentation of the figures is now much clearer and would like to thank the reviewer again for his/her comment.